# Adversarial Examples Might be Avoidable: The Role of Data Concentration in Adversarial Robustness

**Ambar Pal**
ambar@jhu.edu

**Jeremias Sulam**
jsulam1@jhu.edu

**René Vidal**
vidalr@upenn.edu

## Abstract

The susceptibility of modern machine learning classifiers to adversarial examples has motivated theoretical results suggesting that these might be unavoidable. However, these results can be too general to be applicable to natural data distributions. Indeed, humans are quite robust for tasks involving vision. This apparent conflict motivates a deeper dive into the question: Are adversarial examples truly unavoidable? In this work, we theoretically demonstrate that a key property of the data distribution – concentration on small-volume subsets of the input space – determines whether a robust classifier exists. We further demonstrate that, for a data distribution concentrated on a union of low-dimensional linear subspaces, utilizing structure in data naturally leads to classifiers that enjoy data-dependent polyhedral robustness guarantees, improving upon methods for provable certification in certain regimes.

## 1 Introduction, Motivation and Contributions

Research in adversarial learning has shown that traditional neural network based classification models are prone to anomalous behaviour when their inputs are modified by tiny, human-imperceptible perturbations. Such perturbations, called adversarial examples, lead to a large degradation in the accuracy of classifiers [55]. This behavior is problematic when such classification models are deployed in security sensitive applications. Accordingly, researchers have and continue to come up with *defenses* against such adversarial attacks for neural networks.

Such defenses [49, 60, 42, 22] modify the training algorithm, alter the network weights, or employ preprocessing to obtain classifiers that have improved empirical performance against adversarially corrupted inputs. However, many of these defenses have been later broken by new adaptive attacks [1, 8]. This motivated recent impossibility results for adversarial defenses, which aim to show that all defenses admit adversarial examples. While initially such results were shown for specially parameterized data distributions [18], they were subsequently expanded to cover general data distributions on the unit sphere and the unit cube [48], as well as for distributions over more general manifolds [12].

On the other hand, we humans are an example of a classifier capable of very good (albeit imperfect [17]) robust accuracy against $\ell_2$-bounded attacks for natural image classification. Even more, a large body of recent work has constructed *certified* defenses [11, 63, 10, 29, 19, 54] which obtain non-trivial performance guarantees under adversarially perturbed inputs for common datasets like MNIST, CIFAR-10 and ImageNet. This apparent contention between impossibility results and the existence of robust classifiers for natural datasets indicates that the bigger picture is more nuanced, and motivates a closer look at the impossibility results for adversarial examples.

Our first contribution is to show that these results can be circumvented by data distributions whose mass concentrates on small regions of the input space. This naturally leads to the question of whether such a construction is necessary for adversarial robustness. We answer this question in the affirmative, formally proving that a successful defense exists only when the data distribution concentrates on an

37th Conference on Neural Information Processing Systems (NeurIPS 2023).

exponentially small volume of the input space. At the same time, this suggests that exploiting the inherent structure in the data is critical for obtaining classifiers with broader robustness guarantees.

Surprisingly, almost[1] all *certified* defenses do not exploit any structural aspects of the data distribution like concentration or low-dimensionality. Motivated by our theoretical findings, we study the special case of data distributions concentrated near a union of low-dimensional linear subspaces, to create a certified defense for perturbations that go beyond traditional $\ell_p$-norm bounds. We find that simply exploiting the low-dimensional data structure leads to a natural classification algorithm for which we can derive norm-independent polyhedral certificates. We show that our method can certify accurate predictions under adversarial examples with an $\ell_p$ norm larger than what can be certified by applying existing, off-the-shelf methods like randomized smoothing [11]. Thus, we demonstrate the importance of structure in data for both the theory and practice of certified adversarial robustness.

More precisely, we make the following main contributions in this work:

1. We formalize a notion of $(\epsilon, \delta)$-concentration of a probability distribution $q$ in Section 2, which states that $q$ assigns at least $1 - \delta$ mass to a subset of the ambient space having volume $O(\exp(-n\epsilon))$. We show that $(\epsilon, \delta)$-concentration of $q$ is a necessary condition for the existence of any classifier obtaining at most $\delta$ error over $q$, under perturbations of size $\epsilon$.

2. We find that $(\epsilon, \delta)$-concentration is too general to be a sufficient condition for the existence of a robust classifier, and we follow up with a stronger notion of concentration in Section 3 which is sufficient for the existence of robust classifiers. Following this stronger notion, we construct an example of a strongly-concentrated distribution, which circumvents existing impossibility results on the existence of robust classifiers.

3. We then consider a data distribution $q$ concentrated on a union of low-dimensional linear subspaces in Section 4. We construct a classifier for $q$ that is robust to perturbations following threat models more general than $\ell_p$. Our analysis results in polyhedral certified regions whose faces and extreme rays are described by selected points in the training data.

4. We perform empirical evaluations on MNIST in Section 5, demonstrating that our certificates are complementary to existing off-the-shelf approaches like Randomized Smoothing (RS), in the sense that both methods have different strengths. In particular, we demonstrate a region of adversarial perturbations where our method is certifiably robust, but RS is not. We then combine our method with RS to obtain certificates that enjoy the best of both worlds.

## 2 Existence of Robust Classifier Implies Concentration

We will consider a classification problem over $\mathcal{X} \times \mathcal{Y}$ defined by the data distribution $p$ such that $\mathcal{X}$ is bounded and $\mathcal{Y} = \{1, 2, \ldots, K\}$. We let $q_k$ denote the conditional distribution $p_{X|Y=k}$ for class $k \in \mathcal{Y}$. We will assume that the data is normalized, i.e., $\mathcal{X} = B_{\ell_2}(0, 1)$, and the boundary of the domain is far from the data, i.e., for any $x \sim q_k$, an adversarial perturbation of $\ell_2$ norm at most $\epsilon$ does not take $x$ outside the domain $\mathcal{X}$.[2]

We define the robust risk of a classifier $f \colon \mathcal{X} \to \mathcal{Y}$ against an adversary making perturbations whose $\ell_2$ norm is bounded by $\epsilon$ as [3]

$$R(f, \epsilon) = \Pr_{(x,y) \sim p} \left( \exists \bar{x} \in B_{\ell_2}(x, \epsilon) \text{ such that } f(\bar{x}) \neq y \right). \tag{1}$$

We can now define a robust classifier in our setting.

**Definition 2.1** (Robust Classifier). *A classifier $g$ is defined to be $(\epsilon, \delta)$-robust if the robust risk against perturbations with $\ell_2$ norm bounded by $\epsilon$ is at most $\delta$, i.e., if $R(g, \epsilon) \leq \delta$.*

The goal of this section is to show that if our data distribution $p$ admits an $(\epsilon, \delta)$–robust classifier, then $p$ has to be *concentrated*. Intuitively, this means that $p$ assigns a "large" measure to sets of "small" volume. We define this formally now.

---

[1]See Section 6 for more details.
[2]More details in Appendix A.
[3]Note that for $f(\bar{x})$ to be defined, it is implicit that $\bar{x} \in \mathcal{X}$ in (1).

**Definition 2.2** (Concentrated Distribution). *A probability distribution $q$ over a domain $\mathcal{X} \subseteq \mathbb{R}^n$ is said to be $(C, \epsilon, \delta)$-concentrated, if there exists a subset $S \subseteq \mathcal{X}$ such that $q(S) \geq 1 - \delta$ but $\mathrm{Vol}(S) \leq C \exp(-n\epsilon)$. Here, $\mathrm{Vol}$ denotes the standard Lebesgue measure on $\mathbb{R}^n$, and $q(S)$ denotes the measure of $S$ under $q$.*

With the above definitions in place, we are ready to state our first main result.

**Theorem 2.1.** *If there exists an $(\epsilon, \delta)$-robust classifier $f$ for a data distribution $p$, then at least one of the class conditionals $q_1, q_2, \ldots, q_K$, say $q_{\bar{k}}$, must be $(\bar{C}, \epsilon, \delta)$–concentrated. Further, if the classes are balanced, then all the class conditionals are $(C_{\max}, \epsilon, K\delta)$-concentrated. Here, $\bar{C} = \mathrm{Vol}\{x \colon f(x) = \bar{k}\}$, and $C_{\max} = \max_k \mathrm{Vol}\{x \colon f(x) = k\}$ are constants dependent on $f$.*

The proof is a natural application of the Brunn-Minkowski theorem from high-dimensional geometry, essentially using the fact that an $\epsilon$-shrinkage of a high-dimensional set has very small volume. We provide a brief sketch here, deferring the full proof to Appendix A.

*Proof Sketch.* Due to the existence of a robust classifier $f$, *i.e.*, $R(f, \epsilon) \leq \delta$, the first observation is that there must be at least one class which is classified with robust accuracy at least $1 - \delta$. Say this class is $k$, and the set of all points which do not admit an $\epsilon$-adversarial example for class $k$ is $S$. Now, the second step is to show that $S$ has the same measure (under $q_k$) as the $\epsilon$-shrinkage (in the $\ell_2$ norm) of the set of all points classified as class $k$. Finally, the third step involves using the Brunn-Minkowski theorem, to show that this $\epsilon$-shrinkage has a volume $O(\exp(-n\epsilon))$, thus completing the argument. $\qquad\square$

**Discussion on Theorem 2.1.** We pause here to understand some implications of this result.

- Firstly, recall the apparently conflicting conclusions from Section 1 between impossibility results (suggesting that robust classifiers do not exist) and the existence of robust classifiers in practice (such as that of human vision for natural data distributions). Theorem 2.1 shows that whenever a robust classifier exists, the underlying data distribution has to be concentrated. In particular, this suggests that natural distributions corresponding to MNIST, CIFAR and ImageNet might be concentrated. This indicates a resolution to the conflict: concentrated distributions must somehow circumvent existing impossibility results. Indeed, this is precisely what we will show in Section 3.

- Secondly, while our results are derived for the $\ell_2$ norm, it is not very hard to extend this reasoning to general $\ell_p$ norms. In other words, whenever a classifier robust to $\ell_p$-norm perturbations exists, the underlying data distribution must be concentrated.

- Thirdly, Theorem 2.1 has a direct implication towards classifier design. Since we now know that natural image distributions are concentrated, one should design classifiers that are tuned for small-volume regions in the input space. This might be the deeper principle behind the recent success [66] of robust classifiers adapted to $\ell_p$-ball like regions in the input space.

- Finally, the *extent* of concentration implied by Theorem 2.1 depends on the classifier $f$, via the parameters $\epsilon, \delta$ and $\bar{C}$. On one hand, we get *high* concentration when $\epsilon$ is large, $\delta$ is small, and $\bar{C}$ is small. On the other hand, if the distribution $p$ admits an $(\epsilon, \delta)$-robust classifier such that $\bar{C}$ is large (e.g., a constant classifier), then we get *low* concentration via Theorem 2.1. This is not a limitation of our proof technique, but a consequence of the fact that some simple data distributions might be very lowly concentrated, but still admit very robust classifiers, e.g., for a distribution having 95% dogs and 5% cats, the constant classifier which always predicts "dog" is quite robust.

We have thus seen that data concentration is a necessary condition for the existence of a robust classifier. A natural question is whether it is also sufficient. We address this question now.

## 3   Strong Concentration Implies Existence of Robust Classifier

Say our distribution $p$ is such that all the class conditionals $q_1, q_2, \ldots, q_k$ are $(C, \epsilon, \delta)$-concentrated. Is this sufficient for the existence of a robust classifier? The answer is negative, as we have not precluded the case where all of the $q_k$ are concentrated over the same subset $S$ of the ambient space. In other words, it might be possible that there exists a small-volume set $S \subseteq \mathcal{X}$ such that $q_k(S)$ is high for all $k$. This means that whenever a data point lies in $S$, it would be hard to distinguish

which class it came from. In this case, even an accurate classifier might not exist, let alone a robust classifier[4]. To get around such issues, we define a stronger notion of concentration, as follows.

**Definition 3.1** (Strongly Concentrated Distributions)**.** *A distribution $p$ is said to be $(\epsilon, \delta, \gamma)$-strongly-concentrated if each class conditional distribution $q_k$ is concentrated over the set $S_k \subseteq \mathcal{X}$ such that $q_k(S_k) \geq 1 - \delta$, and $q_k \left( \bigcup_{k' \neq k} S_{k'}^{+2\epsilon} \right) \leq \gamma$, where $S^{+\epsilon}$ denotes the $\epsilon$-expansion of the set $S$ in the $\ell_2$ norm, i.e., $S^{+\epsilon} = \{x \colon \exists \bar{x} \in S \text{ such that } \|x - \bar{x}\|_2 \leq \epsilon\}$.*

In essence, Definition 3.1 states that each of the class conditionals are concentrated on subsets of the ambient space, which do not intersect too much with one another[5]. Hence, it is natural to expect that we would be able to construct a robust classifier by exploiting these subsets. Building upon this idea, we are able to show Theorem 3.1:

**Theorem 3.1.** *If the data distribution $p$ is $(\epsilon, \delta, \gamma)$-strongly-concentrated, then there exists an $(\epsilon, \delta + \gamma)$-robust classifier for $p$.*

The basic observation behind this result is that if the conditional distributions $q_k$ had disjoint supports which were well-separated from each other, then one could obtain a robust classifier by predicting the class $k$ on the entire $\epsilon$-expansion of the set $S_k$ where the conditional $q_k$ concentrates, for all $k$. To go beyond this idealized case, we can exploit the strong concentration condition to carefully remove the intersections at the cost of at most $\gamma$ in robust accuracy. We make these arguments more precise in the full proof, deferred to Appendix B, and we pause here to note some implications for existing results.

**Implications for Existing Impossibility Results.** To understand how Theorem 3.1 circumvents the previous impossibility results, consider the setting from [48] where the data domain is the sphere $\mathbb{S}^{n-1} = \{x \in \mathbb{R}^n \colon \|x\|_2 = 1\}$, and we have a binary classification setting with class conditionals $q_1$ and $q_2$. The adversary is allowed to make bounded perturbations w.r.t. the geodesic distance. In this setting, it can be shown (see [48, Theorem 1]) that any classifier admits $\epsilon$-adversarial examples for the minority class (say class 1), with probability at least

$$1 - \alpha_{q_1} \beta \exp \left( -\frac{n-1}{2} \epsilon^2 \right), \tag{2}$$

where $\alpha_{q_1} = \sup_{x \in \mathbb{S}^{n-1}} q_1(x)$ depends on the conditional distribution $q_1$, and $\beta$ is a normalizing constant that depends on the dimension $n$. Note that this result assumes little about the conditional $q_1$. Now, by constructing a strongly-concentrated data distribution over the domain, we will show that the lower bound in (2) becomes vacuous.

**Example 3.1.** *The data domain is the unit sphere $\mathbb{S}^{n-1}$ equipped with the geodesic distance $d$. The label domain is $\{1, 2\}$. $P$ is an arbitrary, but fixed, point lying on $\mathbb{S}^{n-1}$. The conditional density of class 1, i.e., $q_1$ is now defined as*

$$q_1(x) = \begin{cases} \frac{1}{C} \frac{1}{\sin^{n-2} d(x,P)}, & \text{if } d(x, P) \leq 0.1 \\ 0, & \text{otherwise} \end{cases},$$

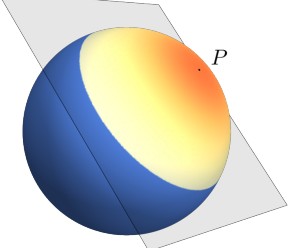

Figure 1: A plot of $q_1$. Redder colors denote a larger density, and the gray plane denotes the robust classifier.

*where $C = 0.1$ is a normalizing constant. The conditional density of class 2 is defined to be uniform over the complement of the support of $q_1$, i.e. $q_2 = \text{Unif}(\{x \in \mathbb{S}^{n-1} \colon d(x, P) > 0.1\})$. Finally, the classes are balanced, i.e., $p_Y(1) = p_Y(2) = 1/2$.*

The data distribution constructed in Example 3.1 makes Eq. (2) vacuous, as the supremum over the density $q_1$ is unbounded. Additionally, the linear classifier defined by the half-space $\{x \colon \langle x, P \rangle \leq \cos(0.1)\}$ is robust (Appendix C provides a derivation of the robust risk, and further comments on generalizing this example). Example 3.1 is plotted for $n = 3$ dimensions in Fig. 1.

---

[4]Recall that classifier not accurate at a point $(x, y)$, i.e., $f(x) \neq y$, is by definition not robust at $x$, as a $v = 0$ perturbation is already sufficient to ensure $f(x + v) \neq y$.

[5]More detailed discussion in Appendix G.

**Compatibility of Theorem 3.1 with existing Negative Results** Thus, we see that strongly concentrated distributions are able to circumvent existing impossibility results on the existence of robust classifiers. However, this does *not* invalidate any existing results. Firstly, measure-concentration-based results [48, 18, 12] provide non-vacuous guarantees given a *sufficiently flat* (not concentrated) data distribution, and hence do not contradict our results. Secondly, our results are existential and do not provide, in general, an algorithm to *construct* a robust classifier given a strongly-concentrated distribution. Hence, we also do not contradict the existing stream of results on the computational hardness of finding robust classifiers [6, 56, 47]. Our positive results are complementary to all such negative results, demonstrating a general class of data distributions where robust classifiers do exist.

For the reminder of this paper, we will look at a specific member of the above class of strongly concentrated data distributions and show how we can practically construct robust classifiers.

## 4 Adversarially Robust Classification on Union of Linear Subspaces

The union of subspaces model has been shown to be very useful in classical computer vision for a wide variety of tasks, which include clustering faces under varying illumination, image segmentation, and video segmentation [57]. Its concise mathematical description often enables the construction and theoretical analysis of algorithms that also perform well in practice. In this section, we will study robust classification on data distributions concentrated on a union of low-dimensional linear subspaces. This data structure will allow us to obtain a non-trivial, practically relevant case where we can show a provable improvement over existing methods for constructing robust classifiers in certain settings. Before delving further, we now provide a simple example (which is illustrated in Fig. 2) demonstrating how distributions concentrated about linear subspaces are concentrated precisely in the sense of Definition 3.1, and therefore allow for the existence of adversarially robust classifiers.

**Example 4.1.** *The data domain is the ball $B_{\ell_\infty}(0,1)$ equipped with the $\ell_2$ distance. The label domain is $\{1,2\}$. Subspace $S_1$ is given by $S_1 = \{x\colon x^\top e_1 = 0\}$, and $S_2$ is given by $S_2 = \{x\colon x^\top e_2 = 0\}$, where $e_1, e_2$ are the standard unit vectors. The conditional densities are defined as*

$$q_1 = \mathrm{Unif}(\{x\colon \|x\|_\infty \le 1, |x^\top e_1| \le e^{-\alpha}/2\}), \text{ and,}$$
$$q_2 = \mathrm{Unif}(\{x\colon \|x\|_\infty \le 1, |x^\top e_2| \le e^{-\alpha}/2\}),$$

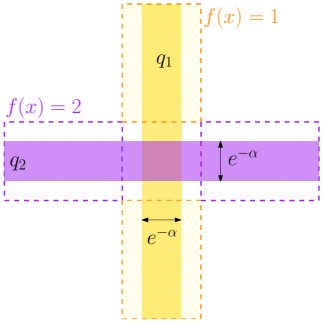

*where $\alpha > 0$ is a large constant. Finally, the classes are balanced, i.e., $p_Y(1) = p_Y(2) = 1/2$. With these parameters, $q_1, q_2$ are both $(0.5, \alpha/n - 1, 0)$-concentrated over their respective supports. Additionally, $p$ is $(\epsilon, 0, e^{-\alpha}/2 + 2\epsilon)$–strongly-concentrated. A robust classifier $f$ can be constructed following the proof of Theorem 3.1, and it obtains a robust accuracy $R(f, \epsilon) \le e^{-\alpha}/2 + 2\epsilon$. See Appendix D for more details.*

Figure 2: A plot of $q_1$ (orange), $q_2$ (violet) and the decision boundaries of $f$ (dashed).

We will now study a specific choice of $p$ that generalizes Example 4.1 and will let us move beyond the above simple binary setting. Recall that we have a classification problem specified by a data distribution $p$ over the data domain $\mathcal{X} \times \mathcal{Y} = B(0,1) \times \{1, 2, \ldots, K\}$. Firstly, the classes are balanced, i.e., $p_Y(k) = 1/K$ for all $k \in \mathcal{Y}$. Secondly, the conditional density, i.e., $q_k = p_{X|Y=k}$, is concentrated on the set $S_k^{+\gamma} \cap \mathcal{X}$, where $S_k$ is a low-dimensional linear subspace, and the superscript denotes an $\gamma$-expansion, for a small $\gamma > 0$.

For the purpose of building our robust classifier, we will assume access to a training dataset of $M$ *clean* data points $(s_1, y_1), (s_2, y_2), \ldots, (s_M, y_M)$, such that, for all $i$, the point $s_i$ lies exactly on one of the $K$ low-dimensional linear subspaces. We will use the notation $\mathbf{S} = [s_1, s_2, \ldots, s_M]$ for the training data matrix and $\mathbf{y} = (y_1, y_2, \ldots, y_M)$ for the training labels. We will assume that $M$ is large enough that every $x \in \cup_k S_k$ can be represented as a linear combination of the columns of $\mathbf{S}$.

Now, the robust classification problem we aim to tackle is to obtain a predictor $g\colon \mathcal{X} \to \mathcal{Y}$ which obtains a low robust risk, with respect to an additive adversary $\mathcal{A}$ that we now define. For any data-point $x \sim p$, $\mathcal{A}$ will be constrained to make an additive perturbation $v$ such that $\mathrm{dist}_{\ell_2}(x + v, \cup_i S_i) \le \epsilon$. In other words, the attacked point can have $\ell_2$ distance at most $\epsilon$ from any of the linear

subspaces $S_1, \ldots, S_k$. Note that $\mathcal{A}$ is more powerful than an $\ell_2$-bounded adversary as the norm of the perturbation $\|v\|_2$ might be large, as $v$ might be parallel to a subspace.

Under such an adversary $\mathcal{A}$, given a (possibly adversarially perturbed) input $x$, it makes sense to try to recover the corresponding point $s$ lying on the union of subspaces, such that $x = s + n$, such that $\|n\|_2 \leq \epsilon$. One way to do this is to represent $s$ as a linear combination of a small number of columns of $\mathbf{S}$, i.e., $x = \mathbf{S}c + n$. This can be formulated as an optimization problem that minimizes the cardinality of $c$, given by $\|c\|_0$, subject to an approximation error constraint. Since such a problem is hard because of the $\ell_0$ pseudo-norm, we relax this to the problem

$$\min_c \|c\|_1 \ \text{s.t.} \ \|x - \mathbf{S}c\|_2 \leq \epsilon. \tag{3}$$

Under a suitable choice of $\lambda$, this problem can be equivalently written as

$$\min_{c,e} \|c\|_1 + \frac{\lambda}{2} \|e\|_2^2 \ \text{s.t.} \ x = \mathbf{S}c + e, \tag{4}$$

for which we can obtain the dual problem given by

$$\max_d \langle x, d \rangle - \frac{1}{2\lambda} \|d\|_2^2 \ \text{s.t.} \ \|\mathbf{S}^\top d\|_\infty \leq 1. \tag{5}$$

Our main observation is to leverage the stability of the set of active constraints of this dual to obtain a robust classifier. One can note that each constraint of Eq. (5) corresponds to one training data point $s_i$ – when the $i^{\text{th}}$ constraint is active at optimality, $s_i$ is being used to reconstruct $x$. Intuitively, one should then somehow use the label $y_i$ while predicting the label for $x$. Indeed, we will show that predicting the majority label among the active $y_i$ leads to a robust classifier.

We will firstly obtain a geometric characterization of the problem in Eq. (5) by viewing it as the evaluation of a projection operator onto a certain convex set, illustrated in Fig. 3. Observe that for $\lambda > 0$, the objective (5) is strongly concave in $d$ and the problem has a unique solution, denoted by $d_\lambda^*(x)$. It is not hard to show that this solution can be obtained by the projection operator

$$d_\lambda^*(x) = \left( \arg\min_d \|\lambda x - d\|_2 \ \text{s.t.} \ \|\mathbf{S}^\top d\|_\infty \leq 1 \right) = \text{Proj}_{K^\circ}(\lambda x), \tag{6}$$

where $K^\circ$ is the polar of the convex hull of $\pm\mathbf{S}$. Denoting $\mathbf{T} = [\mathbf{S}, -\mathbf{S}]$, we can rewrite Problem (6) as $d_\lambda^*(x) = \left( \arg\min_d \|\lambda x - d\|_2 \ \text{sub. to} \ \mathbf{T}^\top d \leq \mathbf{1} \right)$. We now define the set of active constraints as

$$A_\lambda(x) = \{t_i \colon \langle t_i, d_\lambda^*(x) \rangle = 1\}. \tag{7}$$

**Geometry of the Dual** (5). It is illustrated in Fig. 3, where $s_1, s_2$ are two chosen data-points. The blue shaded polytope is $K^\circ$. At $\lambda = \lambda_1$, the point $\lambda_1 x$ lies in the interior of $K^\circ$. Hence, $A_\lambda(x)$ is empty and $\text{supp}(c^*(x))$ is also empty. As $\lambda$ increases, a non-empty support is obtained for the first time at $\lambda = 1/\gamma_{K^\circ}(x)$. For all $\lambda_2 x$ in the red shaded polyhedron, the projection $d_{\lambda_2}^*(x) = \text{Proj}_{K^\circ}(\lambda_2 x)$ lies on the face $F$. As $\lambda$ increases further we reach the green polyhedron. Further increases in $\lambda$ do not change the dual solution, which will always remain at the vertex $d_{\lambda_3}^*(x)$.

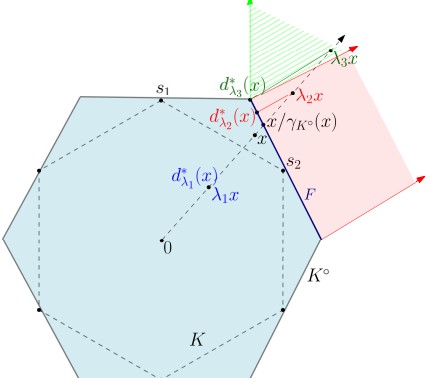

Figure 3: Geometry of the dual problem (5). See description on the left.

Geometrically, $A_\lambda(x)$ identifies the face of $K^\circ$ which contains the projection of $\lambda x$, if $A_\lambda(x)$ is non-empty (otherwise, $\lambda x$ lies inside the polyhedron $K^\circ$). The support of the primal solution, $c^*(x)$, is a subset of $A_\lambda$, i.e. $\text{supp}(c^*(x)) \subseteq A_\lambda(x)$. Note that whenever two points, say $x, x'$, both lie in the same shaded polyhedron (red or green), their projections would lie on the same face of $K^\circ$. We now show this formally, in the main theorem of this section.

**Theorem 4.1.** *The set of active constraints $A_\lambda$ defined in (7) is robust, i.e., $A_\lambda(x') = A_\lambda(x)$ for all $\lambda x' \in C(x)$, where $C(x)$ is the polyhedron defined as*

$$C(x) = F(x) + V(x), \tag{8}$$

*with $F \subseteq K^\circ$ being a facet of the polyhedron $K^\circ$ that $x$ projects to, defined as*

$$F(x) = \left\{ d \; \middle| \; \begin{array}{l} t_i^\top d = 1, \forall t_i \in A_\lambda(x) \\ t_i^\top d < 1, \text{otherwise} \end{array} \right\}, \tag{9}$$

*and $V$ being the cone generated by the constraints active at (i.e., normal to) $F$, defined as*

$$V(x) = \left\{ \sum_{t_i \in A_\lambda(x)} \alpha_i t_i \colon \alpha_i \geq 0, \forall t_i \in A_\lambda(x) \right\}. \tag{10}$$

The proof of Theorem 4.1 utilizes the geometry of the problem and properties of the projection operator, and is presented in Appendix E. We can now use this result to construct a robust classifier:

**Lemma 4.2.** *Define the* dual *classifier as*

$$g_\lambda(x) = \text{AGGREGATE}(\{y_i \colon t_i \in A_\lambda(x)\}), \tag{11}$$

*where AGGREGATE is any deterministic mapping from a set of labels to $\mathcal{Y}$, e.g., MAJORITY. Then, for all $x' \in C(x)$ as defined in Theorem 4.1, $g_\lambda$ is certified to be robust, i.e., $g_\lambda(x') = g_\lambda(x)$.*

**Implications.** Having obtained a certifiably robust classifier $g$, we pause to understand some implications of the theory developed so far. We observe that the certified regions in Theorem 4.1 are not spherical, *i.e.*, the attacker can make additive perturbations having large $\ell_2$ norm but still be unable to change the label predicted by $g$ (see Fig. 4). This is in contrast to the $\ell_2$ bounded certified regions that can be obtained by most existing work on certification schemes, and is a result of modelling data structure while constructing robust classifiers. Importantly, however, note that we do not assume that the attack is restricted to the subspace.

**Connections to Classical Results.** For $\epsilon = 0$, Eq. (3) is known as the primal form of the Basis Pursuit problem, and has been studied under a variety of conditions on $\mathbf{S}$ in the sparse representation and subspace clustering literature [20, 13, 50, 64, 30, 25]. Given an optimal solution $c^*(x)$ of this basis pursuit problem, how can we accurately predict the label $y$? One ideal situation could be that all columns in the support predict the same label, *i.e.*, $y_i$ is identical for all $i \in \text{supp}(c^*(x))$. Indeed, this ideal case is well studied, and is ensured by necessary [25] and sufficient [50, 64, 30] conditions on the geometry of the subspaces $S_1, \ldots, S_K$. Another situation could be that the *majority* of the columns in the support predict the correct label. In this case, we could predict $\texttt{Majority}(\{y_i \colon i \in \text{supp}(c^*(x))\})$ to ensure accurate prediction. Theorem 4.1 allows us to obtain robustness guarantees which work for *any* such aggregation function which can determine a single label from the support. Hence, our results can guarantee robust prediction even when classical conditions are not satisfied. Lastly, note that our Theorem 4.1 shows that the entire active set remains unperturbed – In light of the recent results in [54], this could be relaxed for specific choices of maps acting on the estimated support.

## 5  Experiments

In this section, we will compare our certified defense derived in Section 4 to a popular defense technique called Randomized Smoothing (RS) [11], which can be used to obtain state-of-the-art certified robustness against $\ell_2$ perturbations. RS transforms any given classifier $f \colon \mathcal{X} \to \mathcal{Y}$ to a certifiably robust classifier $g_\sigma^{\text{RS}} \colon \mathcal{X} \to \mathcal{Y}$ by taking a majority vote over inputs perturbed by Gaussian[6] noise $\mathcal{N}(0, \sigma^2 I)$, *i.e.*,

$$g_\sigma^{\text{RS}}(x) = \text{Smooth}_\sigma(f) = \arg\max_{k \in \mathcal{Y}} \Pr_{v \sim \mathcal{N}(0, \sigma^2 I)} (f(x + v) = k). \tag{12}$$

Then, at any point $x$, $g_\sigma^{\text{RS}}$ can be shown to be certifiably robust to $\ell_2$ perturbations of size at least $r^{\text{RS}}(x) = \sigma \Phi^{-1}(p)$ where $p = \max_{k \in \mathcal{Y}} \Pr_{v \sim \mathcal{N}(0, \sigma^2 I)}(f(x + v) = k)$ denotes the maximum probability of any class under Gaussian noise.

It is not immediately obvious how to compare the certificates provided by our method described above and that of RS, since the sets of the space they certify are different. The certified region

---

[6]In reality, the choice of the noise distribution is central to determining the type of certificate one can obtain [41, 63], but Gaussian suffices for our purposes here.

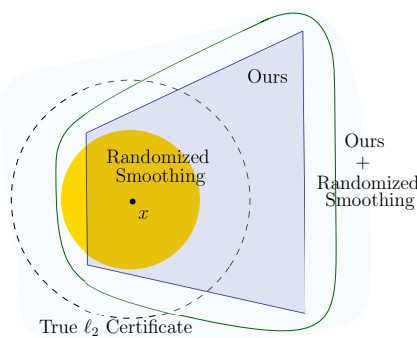

Figure 4: Comparing polyhedral and spherical certificates. Details in text.

obtained by RS, $C^{\mathrm{RS}}(x) = \{\bar{x}: \|x - \bar{x}\|_2 \le r(x)\}$, is a sphere (orange ball in Fig. 4). In contrast, our certificate $C_\lambda(x)$ from Theorem 4.1 is a polyhedron (resp., blue trapezoid), which, in general, is neither contained in $C^{\mathrm{RS}}(x)$, nor a superset of $C^{\mathrm{RS}}(x)$. Additionally, our certificate has no standard notion of *size*, unlike other work on elliptical certificates [14], making a size comparison non-trivial. To overcome these difficulties, we will evaluate two notions of *attack size*: in the first, we will compare the $\ell_2$ norms of successful attacks projected onto our polyhedron, and in the second, we will compare the minimum $\ell_2$ norm required for a successful attack. We will then combine our method with RS to get the best of both worlds, *i.e.*, the green shape in Fig. 4. In the following, we present both these evaluations on the MNIST [28] dataset, with each image normalized to unit $\ell_2$ norm.

**Comparison along Projection on $C_\lambda(x)$.** For the first case, we investigate the question: *Are there perturbations for which our method is certifiably correct, but Randomized Smoothing fails?* For an input point $x$, we can answer this question in the affirmative by obtaining an adversarial example $\bar{x}$ for $g^{\mathrm{RS}}$ such that $\bar{x}$ lies inside our certified set $C_\lambda(x)$. Then, this perturbation $v = x - \bar{x}$ is certified by our method, but has $\ell_2$ norm larger than $r^{\mathrm{RS}}(x)$ (by definition of the RS certificate).

To obtain such adversarial examples, we first train a standard CNN classifier $f$ for MNIST, and then use RS[7] to obtain the classifier $g_\sigma^{\mathrm{RS}}$. Then, for any $(x, y)$, we perform projected gradient descent to obtain $\bar{x} = x^T$ by performing the following steps $T$ times, starting with $x^0 \leftarrow x$:

$$\text{I. } x^t \leftarrow \mathrm{Proj}_{B_{\ell_2}(x,\epsilon)}\Big(x^{t-1} + \eta\nabla_x\mathrm{Loss}(g_\sigma^{\mathrm{RS}}(x^t), y)\Big) \qquad \text{II. } x^t \leftarrow \mathrm{Proj}_{C_\lambda(x)}(x^t) \qquad (13)$$

Unlike the standard PGD attack (step I), the additional projection (step II) is not straightforward, and requires us to solve a quadratic optimization problem, which can be found in Appendix F. We can now evaluate $g_\sigma^{\mathrm{RS}}$ on these perturbations to empirically estimate the robust accuracy over $C_\lambda$, *i.e.*,

$$\mathrm{ProjectionRobustAcc}(\epsilon) = \Pr_{x,y \sim p_{\mathrm{MNIST}}}\Big(\exists \bar{x} \in B_{\ell_2}(x,\epsilon) \cap C_\lambda(x) \text{ such that } g^{\mathrm{RS}}(\bar{x}) \ne y\Big).$$

The results are plotted in Fig. 5, as the dotted curves. We also plot the certified accuracies[7] for comparison, as the solid curves. We see that the accuracy certified by RS drops below random chance (0.1) around $\epsilon = 0.06$ (solid red curve). Similar to other certified defenses, RS certifies only a subset of the true robust accuracy of a classifier in general. This true robust accuracy curve is pointwise upper-bounded by the empirical robust accuracy curve corresponding to any attack, obtained via the steps I, II described earlier (dotted red curve). We then see that even the upper-bound drops below random chance around $\epsilon = 0.4$, suggesting that this might be a large enough attack strength so that an adversary only constrained in $\ell_2$ norm is able to fool a general classifier. However, we are evaluating attacks lying on our certified set and it is still possible to recover the true

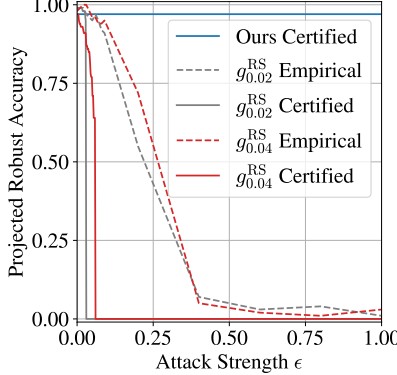

Figure 5: Comparing RS with Our method for adversarial perturbations computed by repeating Steps I, II (13).

---
[7] Further details (e.g., smoothing parameters, certified accuracy computation) are provided in Appendix F.

class (blue solid curve), albeit by our specialized classifier $g_\lambda$ suited to the data structure. Additionally, this suggests that our certified set contains useful class-specific information – this is indeed true, and we present some qualitative examples of images in our certified set in Appendix F. To summarize, we have numerically demonstrated that *exploiting data structure in classifier design leads to certified regions capturing class-relevant regions beyond $\ell_p$-balls*.

**Comparison along $\ell_2$ balls.** For the second case, we ask the question: *Are there perturbations for which RS is certifiably correct, but our method is not?* When an input point $x$ has a large enough RS certificate $r^{\mathrm{RS}}(x) \geq r_0$, some part of the sphere $B_{\ell_2}(x, r^{\mathrm{RS}}(x))$ might lie outside our polyhedral certificate $C_\lambda(x)$ (blue region in Fig. 4). In theory, the minimum $r_0$ required can be computed via an expensive optimization program that we specify in Appendix F. In practice, however, we use a black-box attack [9] to find such perturbations. We provide qualitative examples and additional experiments on CIFAR-10 in Appendix F.

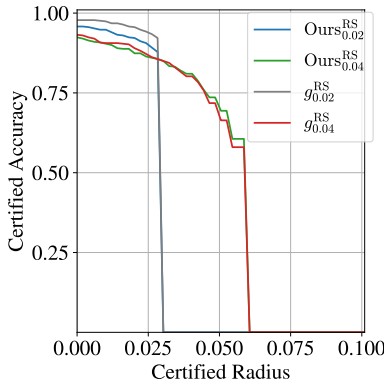

Figure 6: Comparing Certified Accuracy after combining our method with RS.

**Combining Our Method with RS.** We now improve our certified regions using randomized smoothing. For this purpose, we treat our classifier $g_\lambda$ as the base classifier $f$ in (12), to obtain $\mathrm{Smooth}_\sigma(g_\lambda)$ (abbreviated as $\mathrm{Ours}_\sigma^{\mathrm{RS}}$). We then plot the $\ell_2$ certified accuracy[7] [11, Sec 3.2.2] in Fig. 6, where note that, as opposed to Fig. 5, the attacks are *not* constrained to lie on our certificate anymore.

As a final remark, we note that our objective in Fig. 6 was simply to explore RS as a method for obtaining an $\ell_2$ certificate for our method, and we did not tune our method or RS for performance. In particular, we believe that a wide array of tricks developed in the literature for improving RS performance [41, 45, 63] could be employed to improve the curves in Fig. 6. We now discuss our work in the context of existing literature in adversarial robustness, and sparse representation learning.

# 6 Discussion and Related Work

Our proof techniques utilize tools from high-dimensional probability, and have the same flavor as recent impossibility results for adversarial robustness [12, 48, 47]. Our geometric treatment of the dual optimization problem is similar to the literature on sparse-representation [13, 20] and subspace clustering [50, 51, 64, 25], which is concerned with the question of representing a point $x$ by the linear combination of columns of a dictionary $\mathbf{S}$ using sparse coefficients $c$. As mentioned in Section 4, there exist geometric conditions on $\mathbf{S}$ such that all such candidate vectors $c$ are *subspace-preserving*, i.e., for all the indices $i$ in the support of $c$, it can be guaranteed that $s_i$ belongs to the correct subspace. On the other hand, the question of classification of a point $x$ in a union of subspaces given by the columns of $\mathbf{S}$, or subspace classification, has also been studied extensively in classical sparse-representation literature [62, 61, 7, 27, 65, 65, 65, 39, 23, 32]. The predominant approach is to solve an $\ell_1$ minimization problem to obtain coefficients $c$ so that $x = \mathbf{S}c + e$, and then predict the subspace that minimizes the representation error. Various *global* conditions can be imposed on $\mathbf{S}$ to guarantee the success of such an approach [62], and its generalizations [15, 16]. Our work differs from these approaches in that we aim to obtain conditions on perturbations to $x$ that ensure accurate classification, and as such we base our robust classification decision upon properties of solutions of the dual of the $\ell_1$ problem.

Our results are complementary to [44, 5], who obtain a lower bound on the robust risk $R(f, \epsilon)$, in a binary classification setting, in terms of the Wasserstein distance $D$ between the class conditionals $q_0$ and $q_1$, i.e., $R(f, \epsilon) \geq 1 - D(q_0, q_1)$. Using this result, [44, 5] roughly state that the robust risk increases as the class conditionals get closer, i.e., it becomes more likely to sample $x \sim q_0, x' \sim q_1$, such that $\|x - x'\| \leq \epsilon$. In comparison, for two classes, our Theorem 3.1 roughly states that for low robust risk, $q_0$ and $q_1$ should be concentrated on small-volume subsets separated from one another. Thus, the message is similar, while our Theorem 3.1 is derived in a more general multi-class setting. Note that [44, 5] do not explicitly require small-volume subsets, but this is implicit, as $1 - D(q_0, q_1)$

is large due to concentration of measure when $q_0, q_1$ do not concentrate on very small subsets. We provide an analogue of the empirical results of [44, 5] in Appendix F.6.

Our notion of concentration is also related to concentration of a measure $\mu$, as considered in [35, 36, 70], which denotes how much the $\mu$-measure of any set *blows up* after expansion by $\epsilon$. Under this definition, the uniform measure has a high degree of concentration in high dimensions, and this is called the concentration of measure phenomenon. In contrast, our Definition 2.2 of concentrated data distributions can be seen as a *relative* notion of concentration with respect to the uniform measure $\mu$, in that we call a class conditional $q$ concentrated when it assigns a large $q$-measure to a set of very small volume, (*i.e.*, $q(S)$ is high whereas $\mu(S)$ is very low). In essence, Definition 2.2 defines concentration relative to the uniform distribution, whereas [35, 36, 70] define concentration independently. Definition 2.2 is useful in the context of adversarial robustness as it allows us to separate the concentration of data distributions (which is unknown) from the concentration of the uniform measure in high dimensions (for which there is a good understanding). This allows us to derive results in the non-realizable setting, where errors are measured against a probabilistic ground truth label $Y$, which is strictly more general than the realizable setting which requires a deterministic ground truth classifier $f^*$. As a result of this realizable setting, the analysis in [21, 35, 36, 70] needs to assume a non-empty error region $A$ for the learnt classifier $g$ with respect to $f^*$, in order to reason about $\epsilon$-expansions $A^{+\epsilon}$. The results in [35, 36] indicate that the robust risk grows quickly as the volume of $A$ increases. However, humans seem to be a case where the natural accuracy is not perfect (e.g., we might be confused between a 6 and a poorly written 5 in MNIST), yet we seem to be very robust against small $\ell_2$ perturbations. This points to a slack in the analysis in [35, 36], and our work fills this gap by considering $\epsilon$ expansions of a different family of sets.

Finally, our work is also related to recent empirical work obtaining robust classifiers by *denoising* a given input $x$ of any adversarial corruptions, before passing it to a classifier [46, 38]. However, such approaches lack theoretical guarantees, and might be broken using specialized attacks [40]. Similarly, work on improving the robustness of deep network-based classifiers by adversarial training off the data-manifold can be seen as an empirical generalization of our attack model [24, 37, 69, 33]. More generally, it has been studied how adversarial examples relate to the underlying data-manifold [53, 26, 34]. Recent work also studies the robustness of classification using projections onto a single low-dimensional linear subspace [3, 2, 43]. The work in [3] studies an the attack model of bounded $\ell_2, \ell_\infty$ attacks, and they provide robustness certificates by obtaining guarantees on the distortion of a data-point $x$ as it is projected onto a single linear subspace using a projection matrix $\Pi$. In contrast, our work can be seen as projecting a perturbed point onto a union of multiple low-dimensional subspaces. The resultant richer geometry allows us to obtain more general certificates.

## 7   Conclusion and Future Work

To conclude, we studied conditions under which a robust classifier exists for a given classification task. We showed that concentration of the data distribution on small subsets of the input space is necessary for any classifier to be robust to small adversarial perturbations, and that a stronger notion of concentration is sufficient. We then studied a special concentrated data distribution, that of data distributed near low-dimensional linear subspaces. For this special case of our results, we constructed a provably robust classifier, and then experimentally evaluated its benefits w.r.t. known techniques.

For the above special case, we assume access to a *clean* dataset $\mathbf{S}$ lying perfectly on the union of low-dimensional linear subspaces, while in reality one might only have access to noisy samples. In light of existing results on noisy subspace clustering [59], an immediate future direction is to adapt our guarantees to support noise in the training data. Similarly, while the assumption of low-dimensional subspace structure in $\mathbf{S}$ enables us to obtain novel unbounded robustness certificates, real world datasets might not satisfy this structure. We hope to mitigate this limitation by extending our formulation to handle data lying on a general image manifold in the future.

## Acknowledgments and Disclosure of Funding

We would like to thank Amitabh Basu for helpful insights into the optimization formulation for the largest $\ell_2$ ball contained in a polyhedron. This work was supported by DARPA grant HR00112020010,

NSF grants 1934979 and 2212457, and the NSF-Simons Research Collaboration on the Mathematical and Scientific Foundations of Deep Learning (NSF grant 2031985, Simons grant 814201).

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

# A Proof of Theorem 2.1

We will make the technical assumption in Section 2 precise. Recall that all quantities are normalized so that $\mathcal{X}$ is an $\ell_2$ ball of radius 1, i.e., $\mathcal{X} = B_{\ell_2}(0,1)$. Recall that $q_k \colon \mathcal{X} \to \mathbb{R}_{\geq 0}$ denotes the conditional distribution for class $k \in \mathcal{Y}$, having support $Q_k$. We will assume that there is a sufficient gap between the supports and the boundary of the domain, i.e., $Q_k \subseteq B_{\ell_2}(0,c)$, and that the $\ell_2$-adversarial attack has strength $\epsilon \leq c$, for some constant $c$, say $c = 0.1$.

**Theorem 2.1.** *If there exists an $(\epsilon, \delta)$-robust classifier $f$ for a data distribution $p$, then at least one of the class conditionals $q_1, q_2, \ldots, q_K$, say $q_{\bar{k}}$, must be $(\bar{C}, \epsilon, \delta)$–concentrated. Further, if the classes are balanced, then all the class conditionals are $(C_{\max}, \epsilon, K\delta)$-concentrated. Here, $\bar{C} = \mathrm{Vol}\{x \colon f(x) = \bar{k}\}$, and $C_{\max} = \max_k \mathrm{Vol}\{x \colon f(x) = k\}$ are constants dependent on $f$.*

*Proof.* Note that the definition (1) of $(\epsilon, \delta)$-robustness may be rewritten as

$$R(f, \epsilon) = \sum_k p_{X|Y}(X \text{ admits an } \epsilon\text{-adversarial example}|Y = k)p_Y(y = k)$$

$$= \sum_k q_k(\{x \in \mathcal{X} \colon \exists \bar{x} \in B(x, \epsilon) \text{ such that } f(\bar{x}) \neq k\})p_Y(y = k), \qquad (14)$$

where $q(S)$ denotes the $q$-measure of the set $S$. We are given $f$ such that $R(f, \epsilon) \leq \delta$, and we want to find a set $S \subseteq \mathcal{X}$ over which $p$ concentrates.

$$R(f, \epsilon) \leq \delta$$

$$\implies \sum_k q_k(\{x \in \mathcal{X} \colon \exists \bar{x} \in B(x, \epsilon) \cap \mathcal{X} \text{ such that } f(\bar{x}) \neq k\})p_Y(y = k) \leq \delta \qquad (15)$$

Since $\sum_k p_Y(y = k) = 1$, the LHS in (15) is a convex combination, and we have

$$\implies \exists \hat{k} \; q_{\bar{k}}(\{x \in \mathcal{X} \colon \exists \bar{x} \in B(x, \epsilon) \cap \mathcal{X} \text{ such that } f(\bar{x}) \neq \hat{k}\}) \leq \delta, \qquad (16)$$
$$\implies q_{\bar{k}}(U) \leq \delta, \text{ where}$$
$$U = \{x \in \mathcal{X} \colon \exists \bar{x} \in B(x, \epsilon) \cap \mathcal{X} \text{ such that } f(\bar{x}) \neq \hat{k}\}.$$

We will express $q_{\hat{k}}(U)$ in terms of the classification regions of $f$.

Let $A \subseteq \mathcal{X}$ denote the region where the classifier predicts $\hat{k}$, i.e. $A = \{x \in \mathcal{X} \colon f(x) = \hat{k}\}$. Define $A^{-\epsilon}$ to be the set of all points in $A$ at a $\ell_2$ distance atleast $\epsilon$ from the boundary, i.e. $A^{-\epsilon} = \{x \in A \colon B(x, \epsilon) \subseteq A\}$. Now consider any point $x \in \mathcal{X}$. We have the following 4 mutually exclusive, and exhaustive cases:

1. $x$ lies outside $A$, i.e., $x \notin A$. In this case, $x \in U$ trivially as $f(x) \neq k$.

2. $x \in A$, and the entire $\epsilon$-ball around $x$ lies inside $A$, i.e., $B(x, \epsilon) \subseteq A$. In this case, every point in the $\epsilon$-ball around $x$ is classified into class $\hat{k}$, and $x$ has no $\epsilon$-adversarial example for class $\hat{k}$. Note that the set of all these points is $A^{-\epsilon}$ by definition.

3. $x \in A$, and some portion of the $\epsilon$-ball around $x$ lies outside $A$, i.e., $B(x, \epsilon) \not\subseteq A$, and, there is a point $\bar{x} \in B(x, \epsilon) \cap \mathcal{X}$ such that $f(\bar{x}) \neq k$. In this case, $\bar{x}$ is an adversarial example for $x$, and $\bar{x} \in U$.

4. $x \in A$, and some portion of the $\epsilon$-ball around $x$ lies outside $A$, i.e., $B(x, \epsilon) \not\subseteq A$, *but there is no point $\bar{x} \in B(x, \epsilon) \cap \mathcal{X}$ such that $f(\bar{x}) \neq k$.* This can happen only when for all $\bar{x} \in B(x, \epsilon) \setminus A$ we have $\bar{x} \notin \mathcal{X}$. In other words, any $\epsilon$ perturbation that takes $x$ outside $A$, also takes it outside the domain $\mathcal{X}$. This implies that $B(x, \epsilon) \not\subseteq \mathcal{X}$, which in turn means that the distance of $x$ from the boundary of $\mathcal{X}$ is atmost $\epsilon$.

We see that $U$ comprises of the points covered in cases (1) and (3). Recall that we assumed that any point having distance less than $0.1$ to the boundary of $\mathcal{X}$ does not lie in the support of $q_{\bar{k}}$. But all points covered in case (4) lie less than $\epsilon$-close to the boundary, and $\epsilon \leq 0.1$. This implies that $q_{\bar{k}}$

assigns 0 measure to the points covered in case (4). Together, cases (1), (3) and (4) comprise the set $(\mathcal{X} \setminus A) \cup (A \setminus A^{-\epsilon})$. Thus, we have shown

$$q_{\hat{k}}(U) = q_{\hat{k}}(\mathcal{X} \setminus A^{-\epsilon})$$
$$\implies q_{\hat{k}}(A^{-\epsilon}) \geq 1 - \delta. \tag{17}$$

Next, we will show that the set $A^{-\epsilon}$ is exponentially small, by appealing to the Brunn-Minkowski inequality, which states that for compact sets $E, F \subset \mathbb{R}^n$,

$$\mathrm{Vol}(E + F)^{\frac{1}{n}} \geq \mathrm{Vol}(E)^{\frac{1}{n}} + \mathrm{Vol}(F)^{\frac{1}{n}},$$

where Vol is the $n$-dimensional volume in $\mathbb{R}^n$, and $E + F$ denotes the minkowski sum of the sets $E$ and $F$. Applying the inequality with $E = A^{-\epsilon}$ and $F = B_{\ell_2}(0, \epsilon)$, we have

$$\mathrm{Vol}(A^{-\epsilon} + B_{\ell_2}(0, \epsilon))^{\frac{1}{n}} \geq \mathrm{Vol}(A^{-\epsilon})^{\frac{1}{n}} + \mathrm{Vol}(B_{\ell_2}(0, \epsilon))^{\frac{1}{n}}$$
$$\implies \mathrm{Vol}(A)^{\frac{1}{n}} \geq \mathrm{Vol}(A^{-\epsilon})^{\frac{1}{n}} + \epsilon\mathrm{Vol}(B_{\ell_2}(0, 1))^{\frac{1}{n}}$$
$$\implies \mathrm{Vol}(A)^{\frac{1}{n}} \geq \mathrm{Vol}(A^{-\epsilon})^{\frac{1}{n}} + \epsilon\mathrm{Vol}(A)^{\frac{1}{n}}$$
$$\implies \frac{\mathrm{Vol}(A^{-\epsilon})^{\frac{1}{n}}}{\mathrm{Vol}(A)^{\frac{1}{n}}} \leq (1 - \epsilon)$$
$$\implies \mathrm{Vol}(A^{-\epsilon}) \leq \mathrm{Vol}(A)(1 - \epsilon)^n \leq \mathrm{Vol}(A)\exp(-\epsilon n)$$
$$\implies \mathrm{Vol}(A^{-\epsilon}) \leq \bar{C}\exp(-n\epsilon), \tag{18}$$

where $\bar{C} = \mathrm{Vol}(A)$. From (17) and (18), we see that $q_{\bar{k}}$ is $(\bar{C}, \epsilon, \delta)$-concentrated. Additionally, if the classes are balanced, we can use the fact that every term of (15) can be atmost $K\delta$, and apply the above reasoning for each class to conclude that each $q_k$ is $(C_{\max}, \epsilon, K\delta)$-concentrated, for some suitable choice of $C_{\max}$, e.g., $C_{\max} = \max_k \mathrm{Vol}\{x \in \mathcal{X} : f(x) = k\}$. $\qquad\square$

# B  Proof of Theorem 3.1

**Theorem 3.1.** *If the data distribution $p$ is $(\epsilon, \delta, \gamma)$-strongly-concentrated, then there exists an $(\epsilon, \delta + \gamma)$-robust classifier for $p$.*

*Proof.* For each $k \in \{1, 2, \ldots, K\}$, let $S_k$ be the support of the conditional density $q_k$. Recall that $S^{+\epsilon}$ is defined to be the $\epsilon$-expansion of the set $S$. Define $C_k$ to be the $\epsilon$-expanded version of the concentrated region $S_k$ but removing the $\epsilon$-expanded version of all other regions $S_{k'}$, as

$$C_k = \left(S_k^{+\epsilon} \setminus \cup_{k' \neq k} S_{k'}^{+\epsilon}\right) \cap \mathcal{X}.$$

We will use these regions to define our classifier $f : \mathcal{X} \to \{1, 2, \ldots, K\}$ as

$$f(x) = \begin{cases} 1, & \text{if } x \in C_1 \\ 2, & \text{if } x \in C_2 \\ \vdots & \\ K, & \text{if } x \in C_K \\ 1, & \text{otherwise} \end{cases}.$$

We will show that $R(f, \epsilon) \leq \delta + \gamma$, which can be recalled to be

$$R(f, \epsilon) = \sum_k q_k(\{x \in \mathcal{X} : \exists \bar{x} \in B(x, \epsilon) \cap \mathcal{X} \text{ such that } f(\bar{x}) \neq k\})p_Y(y = k).$$

In the above expression, the $q_k$ mass is over the set of all points $x \in \mathcal{X}$ that admit an $\epsilon$-adversarial example for the class $k$, as

$$U_k = \{x \in \mathcal{X} : \exists \bar{x} \in B(x, \epsilon) \cap \mathcal{X} \text{ such that } f(\bar{x}) \neq k\}. \tag{19}$$

Define $C_k^{-\epsilon}$ to be the set of points in $C_k$ at a distance atleast $\epsilon$ from the boundary of $C_k$ as $C_k^{-\epsilon} = \{x \in C_k : B(x, \epsilon) \subseteq C_k\}$. For any point $x \in U_k$, we can find $\bar{x} \in B(x, \epsilon)$ from (19) such that

$\bar{x} \notin C_k$, showing that $x \notin C_k^{-\epsilon}$. Thus, $U_k$ is a subset of the complement of $C_k^{-\epsilon}$, i.e., $U_k \subseteq \mathcal{X} \setminus C_k^{-\epsilon}$, and we have

$$R(f, \epsilon) = \sum_k q_k(U_k) p_Y(y = k) \leq \sum_k (1 - q_k(C_k^{-\epsilon})) p_Y(y = k).$$

Now we will need to show a few properties of the $\epsilon$-contraction. Firstly, for a set $M = N \cap O$, we have $M^{-\epsilon} = N^{-\epsilon} \cap O^{-\epsilon}$, which can be seen as

$$M^{-\epsilon} = \{x \colon x \in M, B(x, \epsilon) \subseteq M\}$$
$$= \{x \colon x \in N, x \in O, B(x, \epsilon) \subseteq N, B(x, \epsilon) \subseteq O\} = N^{-\epsilon} \cap O^{-\epsilon}.$$

Secondly, for a set $M = N^c$, where $c$ denotes complement, we have $M^{-\epsilon} = (N^{+\epsilon})^c$. This can be seen as

$$M^{-\epsilon} = \{x \colon x \in M, B(x, \epsilon) \subseteq M\} = \{x \colon x \notin N, B(x, \epsilon) \subseteq N^c\}$$
$$= \{x \colon x \notin N, \forall x' \in B(x, \epsilon) \; x' \notin N\}$$
$$= \{x \colon \forall x' \in B(x, \epsilon) \; x' \notin N\}$$
$$\implies (M^{-\epsilon})^c = \{x \colon \exists x' \in B(x, \epsilon) \; x' \in N\}$$
$$= N^{+\epsilon}.$$

Thirdly, for a set $M = N \setminus O$, we have $M^{-\epsilon} = (N \cap O^c)^{-\epsilon} = N^{-\epsilon} \cap (O^c)^{-\epsilon}$ by the first property, and then $N^{-\epsilon} \cap (O^c)^{-\epsilon} = N^{-\epsilon} \cap (O^{+\epsilon})^c$ by the second property. This implies

$$(N \setminus O)^{-\epsilon} = N^{-\epsilon} \setminus O^{+\epsilon}.$$

Fourthly, for a set $M = N \cup O$, we have $M^c = N^c \cap O^c$. Taking $\epsilon$-contractions, and applying the first and second properties, we get $M^{+\epsilon} = N^{+\epsilon} \cup O^{+\epsilon}$. Applying the above properties to $C_k^{-\epsilon}$, we have

$$C_k^{-\epsilon} = \left(S_k^{+\epsilon} \setminus \cup_{k' \neq k} S_{k'}^{+\epsilon}\right)^{-\epsilon} \cap \mathcal{X}^{-\epsilon}$$
$$= \left(S_k \setminus \left(\cup_{k' \neq k} S_{k'}^{+\epsilon}\right)^{+\epsilon}\right) \cap \mathcal{X}^{-\epsilon}$$
$$\supseteq \left(S_k \setminus \cup_{k' \neq k} S_{k'}^{+2\epsilon}\right) \cap \mathcal{X}^{-\epsilon}.$$

Recall that $q_k(\mathcal{X}^{-\epsilon}) = q(\mathcal{X}) = 1$ by the support assumption. Hence, we have

$$q_k(C_k^{-\epsilon}) \geq q_k\left(S_k \setminus \cup_{k' \neq k} S_{k'}^{+2\epsilon}\right) = q_k(S_k) - q_k\left(\cup_{k' \neq k} S_{k'}^{+2\epsilon}\right) \geq (1 - \delta) - \gamma$$
$$\implies 1 - q_k(C_k^{-\epsilon}) \leq \delta + \gamma.$$

Finally, as $\sum_k p_Y(y = k) = 1$, we have $R(f, \epsilon) \leq \delta + \gamma$ by convexity. $\qquad \square$

## C   Proofs for Example 3.1

Let $\theta_0 \leq \pi/2$. We will show that the following classifier $f$ is robust in the setting of Example 3.1:

$$f(x) = \begin{cases} 1, & \text{if } x^\top P \geq \theta_0 \\ 2, & \text{otherwise} \end{cases}.$$

Let $C_1 = \{x \in \mathbb{S}^{n-1} \colon x^\top P \geq \theta_0\}$ be the set of points classified into class 1 by $f$. Similarly, let $C_2 = \mathbb{S}^{n-1} \setminus C_1$ be the set of points classified into class 2. The robust risk of $f$ (measured w.r.t. perturbations in the geodesic distance $d$) can be expanded as

$$R_d(f, \epsilon) = 0.5 q_1(d^{+\epsilon}(C_2)) + 0.5 q_2(d^{+\epsilon}(C_1)), \tag{20}$$

where $d^{+\epsilon}(S)$ denotes the $\epsilon$-expansion of the set $S$ under the distance $d$. Let $\epsilon \leq \theta_0$ (otherwise, the first term is $1/2$). Recall that the class conditional $q_1$ is defined as

$$q_1(x) = \begin{cases} \frac{1}{c_\psi} \frac{\psi(d(x,P))}{\sin^{n-2} d(x,P)}, & \text{if } d(x, P) \leq \theta_0 \\ 0, & \text{otherwise} \end{cases},$$

where $c_\psi$ is a normalizing constant which ensures that $q_1$ integrates to 1, *i.e.*, $c_\psi = \int_{\mathbb{S}^{n-1}} q_1$. Then, $q_2$ was defined to be uniform over the complement of the support of $q_1$, i.e. $q_2 = \text{Unif}(\{x \in \mathbb{S}^{n-1} \colon d(x, P) > \theta_0\})$.

We can expand the first term of Eq. (20) as

$$q_1(d^{+\epsilon}(C_2)) = q_1(d^{+\epsilon}(C_2) \setminus C_2) + q_1(C_2)$$
$$= q_1(\{x \colon 0 < d(x, C_2) \leq \epsilon\}) + 0.$$

Now for any $x \notin C_2$, we have $d(x, C_2) = \theta_0 - d(x, P)$. Hence, $\{x \colon 0 < d(x, C_2) \leq \epsilon\} = \{x \colon 0.1 - \epsilon \leq d(x, P) < \theta_0\}$. Let $\theta(x)$ denote the angle that $x$ makes with $P$. Then, the geodesic distance is $d(x, P) = \theta(x)$. The earlier set is the same as the set of all points satisfying $\theta_0 - \epsilon \leq \theta(x) < 1$. This is nothing but the $\epsilon$-base of the hyper-spherical cap having angle $\theta_0$.

We continue,

$$q_1(\{x \colon \theta_0 - \epsilon < d(x, P) \leq \theta_0\}) = q_1(\{x \colon \theta_0 - \epsilon \leq \theta(x) < \theta_0\})$$

With a change of variables, the above can be evaluated as

$$q_1(\{\theta_0 - \epsilon \leq \theta(x) < \theta_0\}) = \frac{1}{c_\psi} \int_{\theta_0 - \epsilon \leq \theta(x) < \theta_0} \frac{\psi(d(x, P))}{\sin(d(x, P))^{n-2}} dx$$
$$= \frac{1}{c_\psi} \int_{\theta_0 - \epsilon}^{\theta_0} \frac{\psi(\theta)}{(\sin \theta)^{n-2}} \mu_{n-2}(\{x \in \mathbb{S}^{n-1} \colon \theta(x) = \theta\}) d\theta,$$

where $\mu_{n-2}$ is the $n-2$-dimensional volume ($n-1$-dimensional surface area). Now, $\mu_{n-2}(\{x \in \mathbb{S}^{n-1} \colon \theta(x) = \theta\})$ is nothing but the volume of an slice of the hypersphere $\mathbb{S}^{n-1}$. This slice is in itself a hypersphere in $\mathbb{R}^{n-1}$, having radius $\sin \theta$. Thus, its volume is the same as the volume of the $\mathbb{S}^{n-2}$, scaled by $(\sin \theta)^{n-2}$. We continue,

$$\int_{\theta_0 - \epsilon}^{\theta_0} \frac{\psi(\theta)}{(\sin \theta)^{n-2}} \mu_{n-2}(\{x \in \mathbb{S}^{n-1} \colon \theta(x) = \theta\}) d\theta = \int_{\theta_0 - \epsilon}^{\theta_0} \frac{\psi(\theta)}{(\sin \theta)^{n-2}} (\sin \theta)^{n-2} d\theta$$
$$= \int_{\theta_0 - \epsilon}^{\theta_0} \psi(\theta) d\theta$$

For illustration, we can take $\psi(\theta) = 1$, giving $q_1(d^{+\epsilon}(C_1)) = \epsilon/C$ from the above integral, with $c_\psi = \int_0^{\theta_0} \psi(\theta) d\theta = \theta_0 = \theta_0$.

We can now follow the same process as above, and expand the Eq. (20) as

$$q_2(d^{+\epsilon}(C_1)) = \mu_{n-1}(\{x \colon \theta_0 \leq \theta(x) \leq \theta_0 + \epsilon\}),$$

where $\mu_{n-1}$ is again the $n-1$-dimensional volume. We use the following formula [31] for the surface area of a hyperspherical cap (valid when $\alpha \leq \pi/2$):

$$\mu_{n-1}(\{x \colon 0 \leq \theta(x) \leq \alpha\}) = m(\alpha) \overset{\text{def}}{=} 0.5 \mu_{n-1}(\mathbb{S}^{n-1}) I_{\sin^2 \alpha}\left(\frac{n-1}{2}, \frac{1}{2}\right),$$

where $I(\cdot, \cdot)$ is the incomplete regularized beta function. Letting $d_{\theta_0} = m(\pi/2) + m(\pi/2) - m(\theta_0)$, we continue,

$$q_2(d^{+\epsilon}(C_1)) = \begin{cases} \frac{1}{d_{\theta_0}}(m(\theta_0 + \epsilon) - m(\theta_0)), & \text{if } \epsilon + \theta_0 \leq \pi/2 \\ \frac{1}{d_{\theta_0}}(m(\pi/2) + m(\pi/2) - m(\theta_0 + \epsilon - \pi/2)), & \text{if } \pi/2 \leq \epsilon + \theta_0 \leq \pi \\ \frac{1}{d_{\theta_0}}, & \text{else} \end{cases}.$$

The risk from class 1 dominates till $\epsilon$ reaches around $\pi/2$, then the risk from class 2 shoots up. We plot the resultant risk in Fig. 7 (left).

Having presented the basic construction, we now comment on how Example 3.1 can be generalized. Firstly, we can vary the support of $q_1$, to have it cover a different fraction of the sphere. For instance, we can set $\theta_0 = \pi/2$ to get a modified $q_1$ as

$$q_1'(x) = \begin{cases} \frac{1}{C} \frac{1}{\sin^{n-2} d(x, P)}, & \text{if } d(x, P) \leq \pi/2 \\ 0, & \text{otherwise} \end{cases},$$

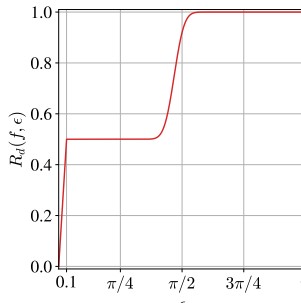 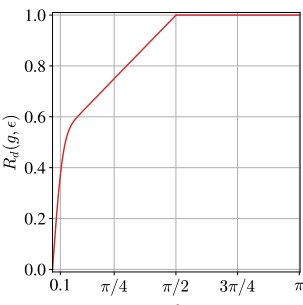 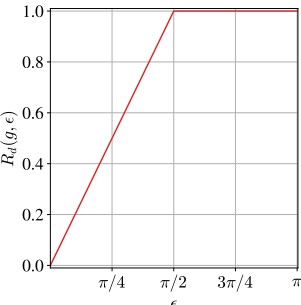

Figure 7: Robust risk for classifiers for concentrated distributions on the sphere $\mathbb{S}^{n-1}$, for dimension $n = 100$. (Left): A plot of $R_d(f, \epsilon)$ w.r.t $p_1$. As $\epsilon$ grows from 0 to 0.1, class 1 is the major contributor to the risk. At $\epsilon = 0.1$, all points in class 1 admit an adversarial perturbation, taking them to class 2. As $\epsilon$ grows further, the risk for class 2 grows slowly, till $\epsilon$ reaches close to $\pi/2$, after which it blows up abruptly due to the high-dimensional fact that most of the mass of the uniform distribution lies near the equator of the sphere. (Middle): A plot of $R_d(g, \epsilon)$ w.r.t. $p_2$. As we make the support of the concentrated class larger, the distribution admits a classifier $g$ with slightly improved robustness for smaller $\epsilon$ values ($\leq 0.1$). (Right): A plot of $R_d(g, \epsilon)$ w.r.t. $p_3$: Finally, as both class conditionals are made concentrated, the classifier $g$ becomes quite robust.

and let $q_2'$ be uniform over the complement of the support of $q_1'$. Along with balanced classes (*i.e.*, $P(Y = 1) = P(Y = 2) = 1/2$), this gives the data distribution $p_2$. The robust classifier would now be given by the half-space $\{x \colon d(x, P) \leq \pi/2\}$. The risk over $p_2$ can be computed as earlier, and is plotted in Fig. 7 (middle).

We can take this example further, and make both $q_1$ and $q_2$ concentrated. This can be done, for instance, by setting $q_1'' = q_1$, and setting $q_2''$ as follows:

$$q_2''(x) = \begin{cases} \frac{1}{C} \frac{1}{\sin^{n-2} d(x, -P)}, & \text{if } d(x, -P) \leq \pi/2 \\ 0, & \text{otherwise} \end{cases},$$

where $-P$ is the antipodal point of $P$. As both class conditionals are now concentrated, the halfspace separating their supports becomes quite robust. This can be seen in the Fig. 7 (right).

Lastly, $\psi$ can be taken to be a rapidly decaying function for even greater concentration and lesser robust risk, e.g., $\psi(\theta) = \exp(-\theta)$.

## D  Proofs for Example 4.1

**Example 4.1.** *The data domain is the ball $B_{\ell_\infty}(0, 1)$ equipped with the $\ell_2$ distance. The label domain is $\{1, 2\}$. Subspace $S_1$ is given by $S_1 = \{x \colon x^\top e_1 = 0\}$, and $S_2$ is given by $S_2 = \{x \colon x^\top e_2 = 0\}$, where $e_1, e_2$ are the standard unit vectors. The conditional densities are defined as*

$$q_1 = \text{Unif}(\{x \colon \|x\|_\infty \leq 1, |x^\top e_1| \leq e^{-\alpha}/2\}), \text{ and,}$$
$$q_2 = \text{Unif}(\{x \colon \|x\|_\infty \leq 1, |x^\top e_2| \leq e^{-\alpha}/2\}),$$

*where $\alpha > 0$ is a large constant. Finally, the classes are balanced, i.e., $p_Y(1) = p_Y(2) = 1/2$. With these parameters, $q_1, q_2$ are both $(0.5, \alpha/n - 1, 0)$-concentrated over their respective supports. Additionally, $p$ is $(\epsilon, 0, e^{-\alpha}/2 + 2\epsilon)$–strongly-concentrated. A robust classifier $f$ can be constructed following the proof of Theorem 3.1, and it obtains a robust accuracy $R(f, \epsilon) \leq e^{-\alpha}/2 + 2\epsilon$. See Appendix D for more details.*

It would be helpful to have Fig. 8 in mind for what follows.

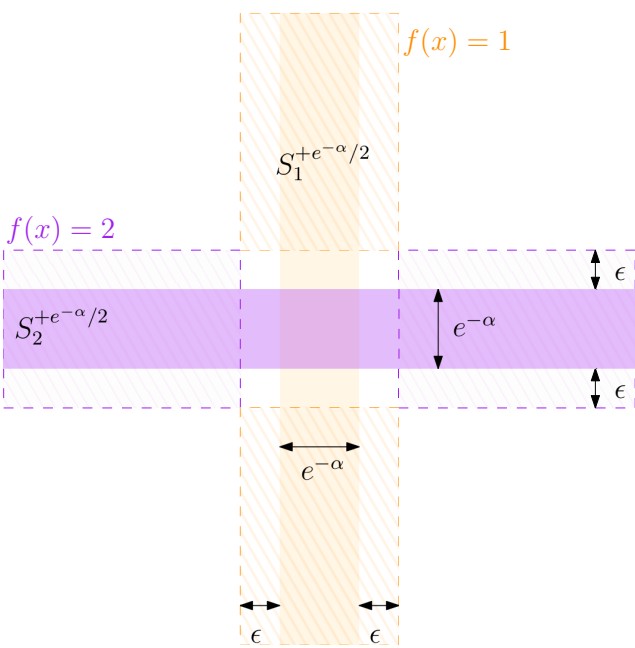

Figure 8: A plot of $q_1$ (orange), $q_2$ (violet) and the decision boundaries of $f$ (dashed).

We will demonstrate that the classifier $f$ illustrated in Fig. 8 has low robust risk. We define $f$ for any $x \in B_\infty(0,1)$, as follows (let $\gamma = e^{-\alpha}$):

$$f(x) = \begin{cases} 1, & \text{if } |x_1| \leq \gamma/2 + \epsilon, |x_2| \geq \gamma/2 + \epsilon \\ 2, & \text{if } |x_2| \leq \gamma/2 + \epsilon, |x_1| \geq \gamma/2 + \epsilon \\ 1, & \text{otherwise} \end{cases}.$$

The reason for splitting the cases for predicting 1 into two subcases, is that we will only need to analyse the first subcase and the second will not contribute to the robust risk.

Defining $U_1$ to be the set of all points $x$ which admit an adversarial example for the class 1, i.e. $U_1 = \{x \colon \exists \bar{x} \text{ such that } \|\bar{x} - x\|_2 \leq \epsilon, f(\bar{x}) = 2\}$. It is clear that

$$U_1 \subseteq \{x \colon |x_1| \geq \gamma/2 \text{ or } |x_2| \leq \gamma/2 + 2\epsilon\}.$$

$U_2$ is defined analogously, and

$$U_2 \subseteq \{x \colon |x_2| \geq \gamma/2 \text{ or } |x_1| \leq \gamma/2 + 2\epsilon\}.$$

Now, $R(f, \epsilon) = 0.5 q_1(U_1) + 0.5 q_2(U_2)$. We look at the first term, and simplify an upper bound:

$$\begin{aligned} q_1(U_1) &\leq q_1(\{x \colon |x_1| \geq \gamma/2 \text{ or } |x_2| \leq \gamma/2 + 2\epsilon) \\ &\leq q_1(\{x \colon |x_1| \geq \gamma/2\}) + q_1(\{x \colon |x_2| \leq \gamma/2 + 2\epsilon) \\ &= 0 + q_1(\{x \colon |x_2| \leq \gamma/2 + 2\epsilon\}). \end{aligned} \tag{21}$$

Now, recall that $q_1 = \text{Unif}(\{x \colon |x_1| \leq \gamma/2\})$, hence the expression in (21) evaluates to

$$\begin{aligned} q_1(\{x \colon |x_2| \leq \gamma/2 + 2\epsilon\}) &= \frac{1}{\text{Vol}(\{x \colon |x_1| \leq \gamma/2\})} \text{Vol}(\{x \colon |x_2| \leq \gamma/2 + 2\epsilon, |x_1| \leq \gamma/2\}) \\ &= \frac{1}{2^{n-1}\gamma} 2^{n-2} \cdot (\gamma + 4\epsilon) \cdot \gamma \\ &= 0.5\gamma + 2\epsilon \end{aligned}$$

The situation for $q_2$ is symmetric, and hence $q_2(U_2) = 0.5(\gamma + 4\epsilon)$. Adding them together, we see that the robust risk is upper bounded by

$$R(f, \epsilon) \leq 0.5(e^{-\alpha} + 4\epsilon)$$

For the concentration parameters, note that the support of $q_1$, *i.e.*, $S_1 = \{x \colon |x_1| \leq \gamma/2, \|x\|_\infty \leq 1\}$ has volume $\gamma 2^{n-1} = 2^{n-1} \exp(-\alpha) = 0.5\, 2^n \exp(-n(\alpha/n)) \leq 0.5 \exp(-n(\alpha/n - 1))$. Hence, $q_1$ is $(0.5, \alpha/n - 1, 0)$ concentrated over $S_1$. Similarly, $q_2$ is $(0.5, \alpha/n - 1, 0)$-concentrated over $S_2 = \{x \colon |x_2| \leq \gamma/2, \|x\|_\infty \leq 1\}$. Finally,

$$
\begin{aligned}
q_1(S_2^{+2\epsilon}) &= q_1(\{\|x\|_\infty \leq 1, \|x_1\| \leq \gamma/2, \|x_2\| \leq \gamma/2 + 2\epsilon\}) \\
&= 2^{n-2}\gamma(\gamma + 4\epsilon)\gamma^{-1}2^{1-n} \\
&= 0.5\gamma + 2\epsilon
\end{aligned}
$$

# E   Proof of Theorem 4.1

**Theorem 4.1.** *The set of active constraints $A_\lambda$ defined in (7) is robust, i.e., $A_\lambda(x') = A_\lambda(x)$ for all $\lambda x' \in C(x)$, where $C(x)$ is the polyhedron defined as*

$$
C(x) = F(x) + V(x), \tag{8}
$$

*with $F \subseteq K^\circ$ being a facet of the polyhedron $K^\circ$ that $x$ projects to, defined as*

$$
F(x) = \left\{ d \;\middle|\; \begin{array}{l} t_i^\top d = 1, \forall t_i \in A_\lambda(x) \\ t_i^\top d < 1, \text{otherwise} \end{array} \right\}, \tag{9}
$$

*and $V$ being the cone generated by the constraints active at (i.e., normal to) $F$, defined as*

$$
V(x) = \left\{ \sum_{t_i \in A_\lambda(x)} \alpha_i t_i \colon \alpha_i \geq 0, \forall t_i \in A_\lambda(x) \right\}. \tag{10}
$$

*Proof.* Let $\lambda x' \in C(x)$ as defined in (8). There exist $f \in F(x), v \in V(x)$ such that $\lambda x' = f + v$. We will show that the projection of $\lambda x'$ onto $F$ is given by $f$. Recall that

$$
A_\lambda(x') = \{t_i \colon \langle t_i, d_\lambda^*(x') \rangle = 1\}, \quad \text{where } d_\lambda^*(x') = \mathrm{Proj}_{K^\circ}(\lambda x'). \tag{22}
$$

Choose any $z \in K^\circ$. Recall that $\mathrm{Proj}_{K^\circ}(\lambda x') = \min_{z \in K^\circ} \|z - \lambda x'\|_2$. Consider the objective,

$$
\|z - \lambda x'\|_2^2 = \|(z - f) - v\|_2^2 = \|z - f\|_2^2 + \|v\|_2^2 - 2\langle z - f, v \rangle. \tag{23}
$$

We will show that $\langle z - f, v \rangle$ is negative. Consider any $t_i \in A_\lambda(x)$, and observe that $\langle z, t_i \rangle \leq 1$, as $z \in K^\circ$. However, from the definition of $F(x)$ recall that $\langle f, t_i \rangle = 1$. This implies that $\langle z - f, t_i \rangle \leq 0$. Using the fact that $v \in V(x)$, we expand the inner product as

$$
\langle z - f, v \rangle = \sum_{t_i \in A_\lambda(x)} \alpha_i \langle z - f, t_i \rangle \leq 0. \tag{24}
$$

Using (24) in (23) to obtain

$$
\|z - \lambda x'\|_2^2 \geq \|z - f\|_2^2 + \|v\|_2^2 \geq 0. \tag{25}
$$

The minimizer above is obtained at $z = f \in C(x)$, and hence we have shown that for all $\lambda x' \in C(x)$, $\mathrm{Proj}_{K^\circ}(\lambda x') = f$. Finally, $d_\lambda^*(x') = \mathrm{Proj}_{K^\circ}(\lambda x') = f \in F(x)$. The theorem statement then follows from the definition of $F(x)$. $\square$

# F   Further Experimental Details and Qualitative Examples

## F.1   Comparison along Projection on $C_\lambda$

In this section, we will provide more details on attacking a classifier with perturbations that lie on our certified set $C_\lambda$.

**Details of the Projection Step II in** (13)    Recall from Theorem 4.1 that $C_\lambda(x)$ is defined as the Minkowski sum of the face $F_\lambda(x)$ (9) and the cone $V_\lambda(x)$ (10). Given an iterate $x^t$, we can compute its projection onto $C_\lambda(x^0)$ by solving the following optimization problem

$$\text{Proj}_{C_\lambda(x^0)}(x^t) = \arg\min_x \|x^t - x\|_2 \text{ s.t. } x \in C_\lambda(x^0)$$

Now every $x \in C_\lambda(x^0)$ can be written as $x = d + v$ where $d \in F(x^0), v \in V(x^0)$. $d$ is such that $t^\top d = 1$ for all $t_i \in A_\lambda(x^0)$, and $t^\top d \leq 1$ otherwise. Then, $v$ is such that $v = \sum_{t_i \in A_\lambda(x^0)} \alpha_i t_i$ for some $\alpha_i \geq 0$. Recall that $T = [t_1, t_2, \ldots, t_{2M}]$ denotes the matrix containing the training data-points as well as their negations. Let the matrix $A_1 \in \mathbb{R}^{d \times |A_\lambda(x^t)|}$ contain all the columns of $T$ in $A_\lambda(x^t)$ and let $A_2 \in \mathbb{R}^{d \times (2M - |A_\lambda(x^t)|)}$ contain all the remaining columns. Then the objective of the optimization problem above can be written as

$$\min_{\alpha, d} \left\| x^t - A_1\alpha - d \right\|_2^2 \text{ s.t. } A_1^\top d = \mathbf{1}, A_2^\top d \leq \mathbf{1}, \alpha \geq \mathbf{0} \tag{26}$$

(26) is a linearly constrained quadratic program, and can be solved by standard optimization tools. In particular, we use the `qp` solver from the python `cvxopt` library.

**Visualization of the Certified Set**    In Fig. 9, we visualize the set $A_\lambda(x)$, *i.e.*, the set of active constraints at the optimal solution of the dual problem, for several $x$ in the MNIST test set. It can be seen that $A_\lambda(x)$ contains images of the same digit as $x$ in most cases. This is expected, as taking the majority label among $A_\lambda(x)$ gives an accuracy of around 97% (solid blue curve in Fig. 5). Note that all these images $t \in A_\lambda(x)$ are also contained in the certified set, $C_\lambda(x)$, which can be seen as $t = t + \mathbf{0}$, with $t \in F_\lambda(x)$, and $\mathbf{0} \in V_\lambda(x)$.

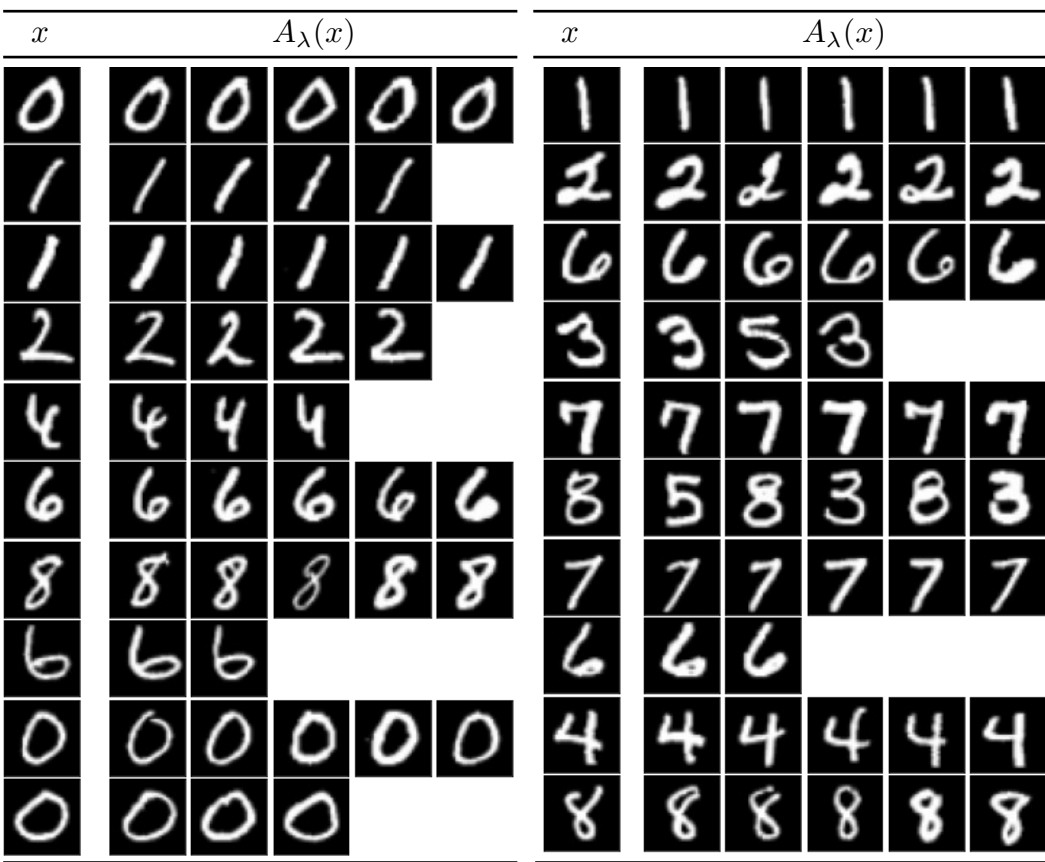

Figure 9: A visualization of the set of active constraints $A_\lambda(x)$ for several $x$. Atmost 5 members of $A_\lambda(x)$ are shown for every $x$.

**Visualization of Attacks restricted to the Certified Set**  In Fig. 10, we visualize the attacks restricted to our certified set $C_\lambda$ computed by (13) for the classifier $g_{0.02}^{\mathrm{RS}}$. We observe that we can identify the correct digit from most of the attacked images, but the randomized smoothing classifier is incorrect at higher $\epsilon$. A small number of these attacked images are close to the actual class decision boundary, and the class is ambiguous. This is expected, both due to the inherent class ambiguity present in some MNIST images, as well as the large $\epsilon$ we are attacking with. For all these images, the prediction of our classifier $g_\lambda$ is certified to be accurate. For comparison, the RS certificate is unable to certify anything beyond $\epsilon \geq 0.06$.

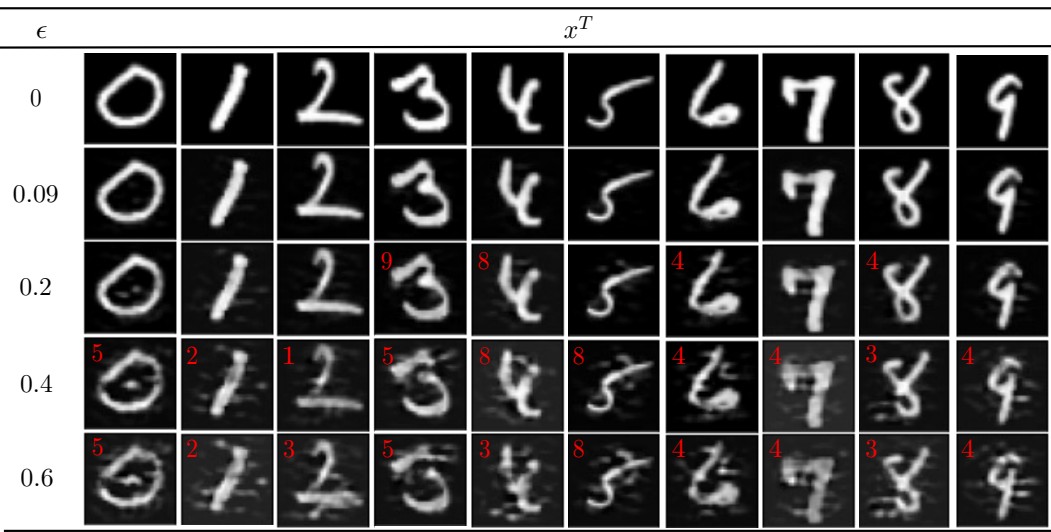

Figure 10: A visualization of attacks on the RS classifier $g_{0.02}^{\mathrm{RS}}$ restricted to our certified set $C_\lambda$ obtained by (13). Different rows plot different attack strengths $\epsilon$. Whenever an image is misclassified, the red annotation on the top left shows the class predicted by $g_{0.02}^{\mathrm{RS}}$.

### F.2  Comparison along $\ell_2$ balls

**Exact Certification for $g_\lambda$**  In this section, we first state and prove Theorem 4.1 that allows us to exactly compute the $\ell_2$ certified radius for our classifier $g_\lambda$ at any point $x$.

**Lemma F.1.** *For all $x' \in C(x)$ as defined in Theorem 4.1, we have $g_\lambda(x') = g_\lambda(x)$. Additionally, for all $\|v\|_2 \leq r_0(x)$ we have $g_\lambda(x + v) = g_\lambda(x)$, where*

$$
\begin{aligned}
r_0(x) = \quad &\min_{\mathbf{u}} \quad -\langle x, \mathbf{u} \rangle \\
&\text{sub. to} \quad \|\mathbf{u}\|_2 = 1, \quad \langle \mathbf{u}, t_i \rangle \leq 0 \; \forall t_i \in A_\lambda(x), \quad \langle \mathbf{u}, v \rangle \leq 1 \; \forall v \in \mathrm{ext}(F(x)),
\end{aligned} \tag{27}
$$

*where $\mathrm{ext}(F(x))$ is the set of extreme points of the polyhedron $F(x)$.*

**Discussion**  Before presenting the proof, let us ponder over the result. We see that (27) involves solving an optimization problem having as many constraints as the number of extreme points of the polyhedron $F(x)$. In general, this can be very large in high dimensions, and hence computationally inefficient to compute. As a result, while Lemma F.1 provides an exact certificate in theory, it is hard to use in practice without further approximations.

*Proof.*  Recall that given a set $A_\lambda(x)$, the polyhedron $C(x)$ is defined in (8) as

$$
C(x) = F(x) + V(x), \tag{28}
$$

where $F(x)$ is a polyhedron, and $V(x)$ is a cone given as the conic hull of $A_\lambda(x)$, *i.e.*, $V(x) = \mathrm{cone}(A_\lambda(x))$. We want to find the size $r(x)$ of the largest $\ell_2$ ball centered at $x$, *i.e.*, $B(x, r(x))$, that can be inscribed within $C(x)$. From convex geometry [4], we know that any polyhedron can be described as an intersection over halfspaces obtained from its polar, $C^\circ$ as

$$
C(x) = \cap_{\mathbf{u} \in C^\circ(x)} H^-(\mathbf{u}, 0), \tag{29}
$$

where $H^-(\mathbf{u}, 0) = \{x \colon \langle \mathbf{u}, x \rangle \leq 0\}$. The distance of $x$ from the boundary of $C(x)$ is the same as the smallest distance of $x$ from any halfspace in (29). However, this distance is simply

$$\text{dist}(x, H^-(\mathbf{u}, 0)) = -\left\langle x, \frac{\mathbf{u}}{\|\mathbf{u}\|_2} \right\rangle. \tag{30}$$

In other words, we can expressing $C(x)$ as the intersection over halfspaces whose normals lie in $C^\circ(x)$, *i.e.*,

$$r(x) = \left( \min_{\mathbf{u}} -\langle x, \mathbf{u} \rangle, \text{ sub. to } \|\mathbf{u}\|_2 = 1, \mathbf{u} \in C^\circ(x) \right). \tag{31}$$

All that is left is to obtain a description of $C^\circ(x)$. This can be done by first expressing $C(x)$ in a standard form, as

$$C(x) = \text{conv}(\text{ext}(F(x))) + \text{cone}(A_\lambda(x)), \tag{32}$$

where $\text{ext}(F(x))$ denotes the extreme points of the polyhedron $F(x)$, $\text{conv}(\cdot)$ denotes the convex hull and $\text{cone}(\cdot)$ denotes the conic hull. Now we can apply a theorem in convex geometry to obtain the polar ([4, Th. 2.79]) as

$$C^\circ(x) = \left\{ \mathbf{u} \colon \begin{array}{l} \langle \mathbf{u}, t_i \rangle \leq 0 \ \forall t_i \in A_\lambda(x) \\ \langle \mathbf{u}, v \rangle \leq 1 \ \forall v \in \text{ext}(F(x)) \end{array} \right\} \tag{33}$$

Combining (31) and (33), we obtain the lemma statement. Finally, we note that as Theorem 4.1 shows that $A_\lambda(x') = A_\lambda(x)$ for all $\lambda x' \in C(x)$, and the classifier $g_\lambda(x)$ is purely a function of $A_\lambda(x)$, we have that $g_\lambda(x') = g_\lambda(x)$ for all $\lambda x' \in C(x)$. $\qquad\square$

**Black Box Attacks on $g_\lambda$** Since our classifier $g_\lambda$ is not explicitly differentiable with respect to its input, we use the HopSkipJump [9] black-box attack to obtain adversarial perturbations that cause a misclassification. The obtained attacked images, and the pertubation magnitude are shown in Fig. 11.

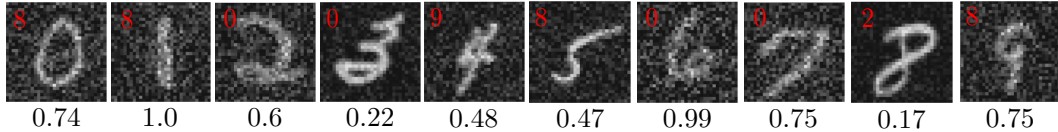

| 0.74 | 1.0 | 0.6 | 0.22 | 0.48 | 0.47 | 0.99 | 0.75 | 0.17 | 0.75 |

Figure 11: A visualization of attacks on our classifier $g_\lambda$ obtained by the HopSkipJump black box attack [9]. The label predicted by $g_\lambda$ is shown as a red annotation on the top left. The $\ell_2$ perturbation magnitude $\epsilon$ is shown at the bottom. Note that the $\epsilon$ values are not nice fractions because they are set by the HopSkipJump attack using a binary search to find the minimum attack strength that causes a misclassification.

## F.3   Details of Randomized Smoothing Certificates

For obtaining the certified accuracy curves using Randomized Smoothing [11] (solid lines in Figs. 5 and 6), we follow the certification and prediction algorithms for RS in [11, Sec 3.2.2]: given a base classifier $f$ and a test image $x$, we use $n_0 = 100$ samples from an isotropic gaussian distribution $\mathcal{N}(0, \sigma)$ to prediction the majority class $g_\sigma^{\text{RS}}(x)$. We then use $n = 100$ samples to estimate the prediction probability $\underline{p_A}$ under $\mathcal{N}(0, \sigma)$ with confidence atleast 0.999. If $\underline{p_A}$ is at least 0.5, we report the certified radius $\sigma \Phi^{-1}(\underline{p_A})$. Otherwise, if $\underline{p_A} < 0.5$, we abstain and return a certified radius of 0.

**Evaluating the Dual Classifier $g_\lambda$** We normalize each image in the MNIST dataset to have unit $\ell_2$ norm, and resize to $32 \times 32$. We pick 10000 images $s_i$ at random from the training set to obtain our data matrix $\mathbf{S} \in \mathbb{R}^{1024 \times 10000}$. We now pick a test point $x$, and then solve the dual problem (5) with $\lambda = 2$, to obtain the dual solution $d_\lambda^*(x)$. This allows us to then obtain the set of active constraints $A_\lambda(x)$ in (7). Then, we use the majority rule as the aggregate function in (11) to obtain the classfication output from the dual classifier $g_\lambda(x)$. We sample 100 and 500 images $x$ uniformly at random from the MNIST test set for generating the curves in Figs. 5 and 6, respectively. All experiments are performed on a NVIDIA GeForce RTX 2080 Ti GPU with 12 GB memory.

### F.4 Discussion of Computational Cost

The computational complexity of obtaining the certificate $C(x)$ in Theorem 4.1 is dominated by solving the linear optimization problem (6) to obtain the set $A_\lambda$, which has $n + M$ variables. It is known that in practice, the cost of solving LPs is much lower than the worst case [52], and it takes us 11.7 seconds on average on a single CPU for each image $x$ without parallel processing.

For comparison to other certified defenses (like Randomized Smoothing), we perform $T = 20$ steps of (13), and each Step II of (13) requires solving a quadratic optimization program, given in (26). This is a linearly constrained quadratic program. In practice it takes us 18.1 seconds on average on a single CPU for each $x$ without parallel processing.

### F.5 Experiments on CIFAR-10

We provide experiments for our method applied to CIFAR-10. To do this, we first embed each CIFAR-10 image into a feature space, designed such that $\ell_2$-bounded perturbations in the feature space, i.e., $\phi(x+v), \|v\|_2 \le \epsilon$, correspond to semantic perturbations in the input space, e.g., distorting the image. Such a feature space is obtained following recent popular work in learning perceptual metrics in vision [67], where the task is precisely to learn a feature representation where $\ell_2$ distances align with human perception. We now use our subspace model on $\phi(X)$, and perform exactly the experiments in Section 5, and make similar observations: we obtain reasonable robust accuracy using our method (Fig. 12 blue lines), and this accuracy is maintained within our certified polyhedra at high $\epsilon$, where existing defenses are not robust.

Though such experiments showing feature space certificates have appeared in the literature [58], the question of whether the perceptual representation is itself susceptible to small adversarial perturbations still remains for future exploration.

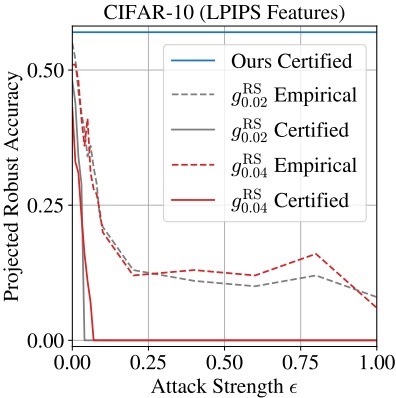 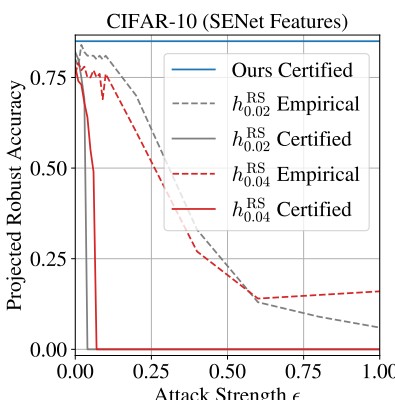

Figure 12: Comparing Randomized Smoothing (RS) with Our Method from Section 4, for adversarial perturbations computed by repeating Steps I, II from Eq. (13). This figure is the counterpart of Fig. 5, where the MNIST dataset has been changed to $\phi(\text{CIFAR-10})$, and the transformation $\phi$ is given by (left) 384-dimensional slice of LPIPS [67] features (i.e., output of the third Conv2D layer) and (right) 128-dimensional input features used by SENet [68]. Observation 1: Our method retains robustness even when the dashed empirical upper bound for robust accuracy for the RS classifier drops to random chance. Observation 2: Our robust accuracy (solid blue line) gets higher from left to right, as the feature space of SENet is closer to a union of subspaces than LPIPS. On the other hand, the left feature space aligns better with human perception than the right feature space.

### F.6 Evaluating Theorem 3.1 on Empirical Distributions

The empirical results in [44, 5] create an empirical distribution $\hat{p}$ by selecting two classes from CIFAR-10, which simply places probability mass $1/N$ on each of the $N$ samples in the dataset. [44, 5] evaluate their lower bounds on $\hat{p}$, and not on the actual real world distribution $p$, which might be arbitrarily complex. Hence, it is unclear whether the trends observed hold for $p$. Nevertheless, the

same experiment can be translated to our setting for any real world dataset, our Theorem 3.1 would then show the existence of a robust classifier for $\hat{p}$. We would need to solve a discrete optimization problem for finding the concentrated sets in Theorem 3.1. Even though valid only for $\hat{p}$, this analysis is interesting, and is reported in Fig. 13.

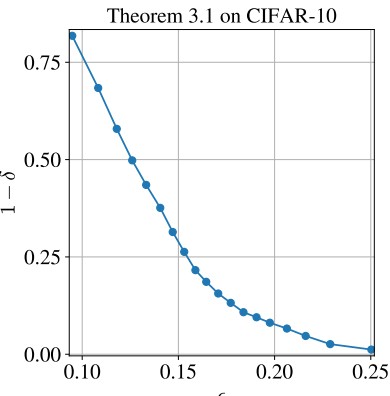

Figure 13: Instantiating Theorem 3.1 for the CIFAR-10 empirical distribution. We construct the empirical distribution $\hat{p}$ which assigns equal probability to each $(x, y)$ pair in the CIFAR-10 test set. This leads to empirical class conditionals $\hat{q}_1, \ldots, \hat{q}_{10}$, corresponding to the 10 classes. We set $\gamma = 0$ in Theorem 3.1, and greedily construct the sets $S_1, \ldots, S_{10}$: sort all images in decreasing order of their distance to their nearest neighbor in the test set, pick the first $m$ images, and then compute the concentration parameters (minimum separation $\epsilon_m$ and mass $\delta_m = 1 - \min_i \hat{q}_i(S_i)$). We then plot $(\epsilon_m, 1 - \delta_m)$ for several choices of $m$. Applying Theorem 3.1, each $(\epsilon_m, \delta_m)$ guarantees the existence of a $(\epsilon_m, \delta_m)$-robust classifier for $\hat{p}$.

# G   Discussion on Strong Concentration

**Are Natural Image Distributions Strongly Concentrated?**   Consider the task of classifying images of dogs vs cats. Any small $\ell_2$ perturbation (say of size $\epsilon = 0.1$) to an image of a dog is not likely to change it to an image of a cat – the true label (or a human decision) does not change with small perturbations. Let $S_{\text{cat}}$ be the set of images that are cats with a high confidence, and similarly for $S_{\text{dog}}$. What we just said was that any image in the $\epsilon$-expansion of $S_{\text{cat}}$ is not likely to be a dog, and vice versa. This is precisely the condition of separation in the definition of strong concentration: $q_{\text{dog}}(S_{\text{cat}}^{+\epsilon}) \leq \gamma$, and $q_{\text{cat}}(S_{dog}^{+\epsilon}) \leq \gamma$, for a small $\gamma$.

Now, despite the above, the task of practically classifying an image into a dog or a cat is not trivial at all, as the sets $S_{\text{dog}}$ and $S_{\text{cat}}$ might be very complex, and hence it might be computationally hard to find a predictor (neural network, or any other classifier) for distinguishing $S_{\text{dog}}$ and $S_{\text{cat}}$. Using modern deep learning, one is able to learn an approximation to these sets that leads to small standard error, but this approximation is still bad enough that the learned predictor is very susceptible to adversarial examples. This is to say that task of robustly classifying an image into dog or cat is even harder.

**Tightness of Strong Concentration**   In Definition 3.1, we are trying to obtain a sufficient condition for the existence of a robust classifier. There are two parts to Definition 3.1: the concentration, and the separation conditions. The first is essential, as Theorem 2.1 asserts that whenever a robust classifier exists for a data distribution, the class conditionals are concentrated (i.e., it is necessary).

As we argued above, the separation condition is not very strong, and should be satisfied by image distributions like MNIST, CIFAR-10 and ImageNet. Let us walk through this condition to see if it can be further relaxed. As we note at the start of Section 3, if all the class conditionals were to be concentrated on subsets $S_k$ that have high intersection among each other, then even benign classification would be hard (i.e., benign risk will be high), let alone robust classification. So, even for the existence of a good benign classifier, it is essential for $S_k \cap S_{k'}$ to be close to empty, for all $k \neq k'$.

Now, even if $S_k \cap S_{k'}$ were almost empty, for a robust classifier, we care about the $\epsilon$-expansions of these sets to not intersect. In other words, we do not want an $\epsilon$-perturbed cat to look like a dog. Hence, for the existence of a good robust classifier, $S_k \cap S_{k'}^{+\epsilon}$ should be close to empty, for all $k \neq k'$. This can be generalized in measure terms, to require $q_k(S_{k'}^{+\epsilon}) \leq \gamma$, for a small $\gamma$. Upto this, all these conditions are essential for obtaining a robust classifier.

Now, note that in Definition 3.1, the expansion is taken for $2\epsilon$, instead of $\epsilon$. This is where our proof technique for Theorem 3.1 incurs a slack of $\epsilon$, and we believe a different approach for constructing the robust classifier might be able to reduce the expansion requirement to $\epsilon$. This could be an interesting future avenue for relaxing Definition 3.1 and thereby strengthening Theorem 3.1.

