# OpenReview forum: "Adversarial Examples Might be Avoidable: The Role of Data Concentration in Adversarial Robustness"
_NeurIPS.cc/2023/Conference — NeurIPS 2023 poster_

### Official Review · Reviewer_Ui3o · 2023-06-27

**Soundness:** 3 good
**Presentation:** 3 good
**Contribution:** 2 fair
**Rating:** 5
**Confidence:** 4

**Summary:**

The paper studies the problem of adversarial robustness in multi-class classification. The authors show that a necessary condition for the existence of an adversarially robust classifier is the condition that at least one of the class conditional ditributions is "concentrated" in the sense that most of the probability mass is concentrated on a set of very small volume. The condition of concentration however is not sufficient for the existence of a robust classifier, because one also needs the class conditional ditributions to be well-separated.

The authors then study adversarial robustness for the special case of data distributions lying on a low-dimensional linear subspace. For this case, they show that robustness at any data point can be certified in a polyhedron that depends on the subspace representation of the point. Using experiments on MNIST dataset, they show that the polyhedron certificate can be complementary to the spherical certificate one can get with randomized smoothing.

**Strengths:**

- Necessary condition: The theorem showing the necessity of concentration for robustness results from a simple, but nice application of the Brunn-Minkowski inequality. I think this condition is novel, and complements existing results showing concentration of measure is bad for robustness
- Polyhedral certificate: The theorem showing the polyhedral certificate for robustness appears non-trivial and interesting. The certificate obtained is complementary to the spherical certificates of randomized smoothing.
- Writing: The paper is well-written and clear for the most part.

**Weaknesses:**

- Definition 3.1: I think the definition of strongly concentrated distributions is a misnomer. The critical assumption in this definition is in line 125 which assumes minimal overlap between the approximate supports of the class conditional distributions. Hence, this definition has more to do with "well-separatedness" of class conditional distributions, and in my opinion, has very little relation to concentration.
- Novelty of the sufficient condition: If concentration on small volume sets is not sufficient for robustness, what is? As I state in the previous point, it is some notion of separation between class conditional distributions. Several papers already study such conditions. For example, [this](http://proceedings.mlr.press/v119/pydi20a/pydi20a.pdf) and [this](https://proceedings.neurips.cc/paper/2019/file/02bf86214e264535e3412283e817deaa-Paper.pdf) paper show that if the class-conditional distributions are well-separated in an optimal-transport distance, then there exists a robust classifier. It is also shown in these papers that such a separation condition holds empirically on real-world datasets.
- The low-dimensional linear subspace assumption: First, I think this is a very strong assumption that real world datasets probably don't satisfy. Second, the motivation for studying distributions of this form appears weak, as one can imagine many other types of distributions that satisfy the necessary and sufficient conditions of sections 2 and 3 but are not supported on linear subspaces. Case in point is Example 3.1 in this very paper.
- The utility of polyhedral certificate: Intuitively, one can imagine many "benevolent perturbations" to input data that do not affect the accuracy of a classifier. Spherical certificates for robustness make sense because the guarantee holds for any possible direction of the adversarial perturbation. A polyhedral certificate is not a "worst-case" certificate in this sense.

**Questions:**

- For $\sigma = 0.04$ (red and green curves) Figure 6 seems to show that the the paper's method gives a worse certificate than randomized smoothing. Is this a typo?
- Is it possible to find the radius of the largest $\ell_p$ ball centered at x that fits inside C(x)? This will give a data-dependent spherical certificate at each point, which will be interesting.

Minor suggestions:
- I think it is better to show the dependence on dimension for the constants $\epsilon, \delta$ in the main theorems of sections 2 and 3.
- The connections of the main theorem of section 4 to subspace clustering literature can probably be presented much earlier. Currently, the paper only brings up this connection at the end of section 4.

**Limitations:**

Yes.

---

> ### Author Rebuttal · Authors · 2023-08-10
>
> Thank you for finding our proof techniques and our polyhedral certificates novel, and the paper well written. We address your questions and concerns below:
>
> > Q2: Is it possible to find the radius of the largest ball around x that fits inside C(x)?
>
> Good question. While a closed form solution is not possible in general, we already provided an optimization problem that obtains this radius in the Appendix F.1. This problem is computationally expensive to solve exactly, and it might be possible to obtain an efficient approximation (like other related work on exact certificates [A]). This is currently mentioned on L345 in the main paper, but we will make it more prominent.
>
> > Q1: Typo in Fig. 6?
>
> There is no typo, $\sigma$ is a hyper parameter for RS - our method performs best at $\sigma = 0.02$, whereas the baseline works best at $\sigma = 0.04$. As L351 mentions, the differences are really slight between the curves, as the $x$-axis is zoomed in.
>
> ---
>
> **On Definition 3.1**
> Sec 3 obtains a sufficient condition for the existence of a robust classifier. There are two parts to Def 3.1: the concentration, and the separation parts. The separation condition requires existence of sets $S_i$ such that $q_i(S_j^{2\epsilon})$ is small for $i \neq j$. Without further restrictions, the empty sets are trivially separated. The missing ingredient for ensuring robustness is that the measure $q_i(S_i)$ should be large enough. Now both these conditions cannot be true without the volume of $S_i$ being small enough (as the volume of the expansion blows up otherwise): this is precisely our concentration part. Thus, concentration is important for any sufficient condition. In fact, please see the [global response][1] for a detailed discussion on why Def 3.1 is essentially the minimum required for any sufficient condition for robustness.
>
> **Comparison to [1,2]** Our results are complementary to [1,2], and are derived in a more general multi-class setting:
>
> The theoretical results in [1,2] obtain a nice lower bound in *a binary classification setting* on the adversarial risk $R$ in terms of the wasserstein distance $D$ between the class conditionals $q_0$ and $q_1$, i.e., $R(f, \epsilon) \geq 1 - D_\epsilon(q_0, q_1)$. Our theoretical results Thm 2.1, Thm 3.1 are derived in the multi-class setting, and a priori it is unclear how to extend the results in [1,2] to the multiclass setting. Nevertheless, [1,2] roughly states that the lower bound increases as it becomes more likely to find $x \sim q_0$, $x' \sim q_1$ such that $||x - x'|| \leq 2\epsilon$, i.e., as the class conditionals get *closer*. In comparison, for two classes, our Theorem 3.1 roughly states that $q_0$, $q_1$ should be concentrated on small subsets $2\epsilon$ separated from one another. Thus, both our Thm 3.1 and [1, 2] have a similar message, while Thm 3.1 is valid in a more general multi class setting. Finally, note that the *small subsets* part is not explicitly mentioned in [1, 2], but is implicit, as $1 - D_\epsilon(q_0, q_1)$ is large due to concentration of measure when $q_0,q_1$ do not concentrate on very small subsets.
>
> The empirical results in [1,2] create an *empirical distribution* $\hat p$ by selecting *two* classes from real-world datasets (CIFAR-10, Fashion MNIST), which simply places probability mass $1/N$ on each of the $N$ samples in the dataset. [1,2] evaluate their lower bounds on $\hat p$, and not on the actual real world distribution $p$ -- which might be arbitrarily complex -- and hence we would not say that "their condition holds empirically for real world datasets", as the reviewer mentions. In fact, the same experiment can be translated to our setting for any real world dataset, our Thm 3.1 would then show the existence of a robust classifier for $\hat p$. Just like [1,2] need to solve a discrete optimization problem to evaluate their lower bound on $\hat p$, we would need to solve a discrete problem for finding the concentrated sets $S$ in Thm 3.1. Even though valid only for $\hat p$, these analyses are interesting, and have been added to the [global response][1].
>
> Thanks for pointing out these references, we will add them to the related work section.
>
> **On Polyhedral Certificates**
> As the reviewer mentions, for any image $x$ there are some benevolent directions $v$ that do not matter for a human (e.g. changing the background), and other directions that do. Spherical certificates treat every direction as the same, and hence cannot capture the above asymmetry. Polyhedral certificates provide a way to move beyond such restrictions, are a step towards certificates aligned with human perception, in our opinion. Additionally, one can always find the largest ball inside a polyhedron (Q2 above) to obtain a spherical certificate from a polyhedral one.
>
> **On Distributional Assumptions in Sec 4**
> Sec. 4 deals with a very specific instantiation of our general theory, for data distributions supported *near* low dimensional linear subspaces, an assumption suited for several real-world tasks including cancer subtyping [B], hyper spectral image segmentation [C] and face recognition. Studying such assumptions is useful, as we are able to obtain novel polyhedral certificates that move beyond the $\ell_p$ bounds studied in the certified robustness literature. However, any assumption has datasets that might not satisfy it, like CIFAR-10. Nevertheless, our theory can still be used for such datasets, and we provide experiments and thoughts in the [global response][1] showing a possible path.
>
> **Minor Suggestions** Thanks for the suggestions, we will reconsider placement of the connections to subspace clustering literature, and mention the dependence on the constants.
>
> [A]:Wang et.al., Beta-CROWN, NeurIPS 2021
> [B]:McWilliams et.al., Subspace clustering of high-dimensional data
> [C]:Zhang et.al., Spectral–spatial sparse subspace clustering for hyperspectral remote sensing images
>
> [1]: https://openreview.net/forum?id=JDoA6admhv&noteId=hpTXK3MMPW

---

### Official Review · Reviewer_kZpV · 2023-07-06

**Soundness:** 3 good
**Presentation:** 3 good
**Contribution:** 3 good
**Rating:** 6
**Confidence:** 3

**Summary:**

This paper gives a theoretical insight into the existence of robust classifiers for avoiding adversarial examples, which may be a potential direction for bridging the gap between human perception and machine intelligence. Specifically, this paper reveals the sufficiency and necessity of data concentration for robust classification. The authors conclude that concentration on small-volume subsets of the input space is the key to avoiding adversarial examples for machine learning. In two experiments, the proposed theoretical results are shown to be effective in improving projected robust accuracy and certified accuracy.

**Strengths:**

- The research problem is quite interesting, and is significant for the development of machine robustness and safety.
- This paper is well-organized. The analyses are rigorous and solid.

**Weaknesses:**

- The main theoretical results are based on $\ell_2$-normed adversarial training, whether the conclusions are still applicable on $\ell_\inf$ norm is unknown.

- The empirical results are insufficient. The experiments are only conducted on every small dataset, whether the theoretical conclusion still holds on larger natural datasets is unknown.

- the conclusion related to subspace training is similar to some existing adversarial training methods, such as [1], [2], which shows that adversarial training on subspace can help alleviate overfitting and improve generalization performance. Is there any difference between these works?
[1] Li et al., Subspace adversarial training, CVPR 2022.
[2] Li et al., On the adversarial robustness of subspace learning, TIP 2020.

- Another question is that whether the subspace preserve training could be applied to transferred adversarial attacks. Could you make several discussions to justify this?

**Questions:**

Please refer to weaknesses.

**Limitations:**

Limitations are well-discussed. No potential negative societal impact is found.

---

> ### Author Rebuttal · Authors · 2023-08-09
>
> Thank you for finding our paper well organized and our theory solid. We address your questions below:
>
> > The empirical results are insufficient. The experiments are only conducted on a small dataset, whether the theoretical conclusion still holds on larger natural datasets is unknown.
>
> **CIFAR-10 Experiments**
>
> We conducted additional experiments on CIFAR-10 (albeit with some important modifications mentioned below), and obtain similar conclusions as in the paper. The figures for these new results can be found in the PDF attached with the [global response][9]. Details:
>
> - Firstly, we note that our theoretical conclusions (Thm 2.1, 3.1) hold for general data distributions. In the paper, we instantiated our results and provided experiments for distributions supported near a union of low dimensional linear subspaces.
>
> - For our experiments on CIFAR-10, we first embed each CIFAR-10 image $x$ into a feature space $\phi(x)$, designed such that $\ell_2$-bounded perturbations in the feature space, i.e., $\phi(x) + v$, $||v||_2 \leq \epsilon$, correspond to *semantic* perturbations in the input space, e.g. distorting the image. Such a feature space $\phi$ is obtained following recent popular work in learning perceptual metrics in vision [A], where the task is precisely to learn a feature representation where $\ell_2$ distances align with human perception. Once everything is projected to this feature space, we can utilize our subspace model, and perform exactly the experiments in Sec 5. This style of experiments showing certified robustness after projection to a feature space has appeared in prior literature [B], however the question of whether the perceptual representation $\phi$ is itself susceptible to small adversarial perturbations remains, and should be a study of future work.
>
> - We find that the results are similar to those observed in the paper: we are able to obtain a non-trivial robust accuracy using our method, and this accuracy is maintained at high epsilons within our certified polyhedra.
>
> > The main theoretical results are based on $\ell_2$-normed adversarial training, whether the conclusions are still applicable on $\ell_\infty$ norm is unknown.
>
> ---
>
> **Extensions to $ \ell_\infty $**
> Extensions of our methods to $\ell_\infty$ perturbations is very much possible, and is a subject of ongoing work. We comment with respect to each of our results, in increasing order of difficulty of extending to $\ell_\infty$:
>
> - For the sufficient condition in Sec 3 (Theorem 3.1), our proof technique is in fact quite general, and extends directly to other $\ell_p$ distances, including $\ell_\infty$.
>
> - For the necessary condition in Sec 2 (Theorem 2.1), the general style of the argument mentioned in the proof sketch (L91) still works, the main technical challenge is to show that the $\epsilon$-shrinkage under the $\ell_\infty$ distance has a small volume. We can no longer utilize Brunn-Minkowski, and have to resort to more sophisticated tools from high dimensional geometry.
>
> - For our algorithmic contribution in Sec 4, we studied Problem (3), with a constraint on the $\ell_2$ norm of the residual $x - Sc$. This constraint would be replaced with the $\ell_\infty$ norm. Now the analysis after equation (4) would look quite different with this change, and the associated certificates are also likely to change.
>
> ---
>
> **Other Questions**
> > Comparison to [1] Li et al., Subspace adversarial training, CVPR 2022. and [2] Li et al., On the adversarial robustness of subspace learning, TIP 2020.
>
> We believe [1, 2] are only tangentially related to our work, as we detail below:
> [1] deals with training in a low dimensional subspace of the *weight space* whereas our Sec 4. considers a low dimensional subspace assumption in the *input space*. Further, our certificates are derived against a general $\ell_2$-bounded adversary, who is *not* restricted to a subspace.
> [2] studies the problem of making Frobenius-Norm-bounded perturbations to a data matrix $X$ to obtain $X'$ such that the distance between the subspaces learnt using PCA, i.e. $\theta({\rm PCA}(X), {\rm PCA}(X'))$ is maximized, for some notion of distance $\theta$. This is a regression problem, whereas our paper studies a classification problem, with very different attack models.
>
> > Another question is that whether the subspace [preserving] training could be applied to transferred adversarial attacks. Could you make several discussions to justify this?
>
> We are unsure whether we understand the question. Could the reviewer please clarify what they mean by subspace adversarial training (presumably [1]) "applied to transferred adversarial attacks"?
>
>
> [A]: Zhang et. al., The Unreasonable Effectiveness of Deep Features as a Perceptual Metric, CVPR 2018
>
> [B]: Voracek et. al., Provably Adversarially Robust Nearest Prototype Classifiers, ICML 2022
>
> [9]: https://openreview.net/attachment?id=hpTXK3MMPW&name=pdf

---

### Official Review · Reviewer_eEv4 · 2023-07-06

**Soundness:** 3 good
**Presentation:** 3 good
**Contribution:** 2 fair
**Rating:** 8
**Confidence:** 4

**Summary:**

The paper shows that adversarial examples can be avoided allowing for robust classification as long as the data distribution obeys certain properties. In particular, they show that the the input data distribution needs to be "well-concentrated" and not overlap too much in order to guarantee the existence of a robust classifier with provable certificates. They support their theoretical guarantees with experimental results on MNIST.

**Strengths:**

1. The problem of adversarial robustness is extremely important and impactful. Further, in the face of hardness results and well known existence of adversarial examples, this is a difficult problem to tackle
2. The authors formalize the notion of "well-concentrated" distributions very well. I think the idea of distribution-specific robustness is well motivated and the right way to approach this problem.
3. I think the paper is well written and the mathematical arguments look non-trivial and sound.

**Weaknesses:**

1. Since this work is primarily a theoretical contribution, I consider this a minor point - I would like to see experimental results done on a few settings larger than MNIST. (say Cifar-10, TinyImagenet)
2. My main concern with the paper is that Definition 3.1 is possibly too strong. It almost seems like the definition is equivalent to creating a robust classifier since it ensures that the different classes are well separated and well concentrated. Is there some way to interpolate between Definition 2.2 and 3.1 which is the minimum assumptions required to guarantee the existence of a robust classifier?

**Questions:**

1. In line 229, do you mean strongly concave? Unless you transform the problem to a minimization problem.
2. The assumption that adversarial perturbations do not take $x$ outside the domain $\mathcal{X}$ seems somewhat unreasonable. Is this really necessary? I agree that this is needed for $f(\bar{x})$ to be well defined, but isn't it even easier to guarantee robustness if it is not defined?
3. The notion of requiring the data distribution to concentrate on an exponentially small volume of the input space looks closely related to the hardness results from Wang et al. "Attack of the Tails: Yes, You Really Can Backdoor Federated Learning". Their results show that adversariabl robustness in FL is provably hard which is the opposite of this work. However, I feel they are somewhat related. Can you comment?
4. In (1), can the definition be changed to $f(\bar{x}) \neq f(x)$? This way robustness and accuracy can be disentangled. It seems somewhat unintuitive that an inaccurate classifier cannot be robust. I would expect that the $0$ classifier is the most robust.
5. I have perhaps missed this, but I couldn't find any discussion of the computational cost of certifying robustness using this approach. I would like to see the dependence on $d$.
6. In Figure 5, the "Ours Certified" line is 1.0 for all $\epsilon \in [0, 1]$. However, the certified region should not be symmetric around $x$, so this doesn't make sense to me.

**Limitations:**

The authors discuss some of the limitations of their work in Section 6. But I would like them to consider some of my questions above and include those as well if they are relevant.

---

> ### Author Rebuttal · Authors · 2023-08-09
>
> Thank you for finding our paper well written, and our theory interesting. We address your comments below:
>
> > My main concern is that Def 3.1 is possibly too strong. Is there some way to [relax it]?
>
> Great question. In the [global response][9], we explain why Def 3.1 is not strong, is satisfied by real world distributions, and is $\epsilon$-close to the minimum set of conditions needed. We will add this discussion to the paper.
>
> > Empirical Results on CIFAR-10?
>
> Please see the [global response][9] for empirical results on CIFAR-10.
>
> ---
>
> > Q4: Can Eq. (1) be changed to $f(\bar x) \neq f(x)$?
>
> The short answer is no, the definition should not be changed. Let us expand further:
>
> Let us call the alternative definition $f(\bar x) \neq f(x)$ to be *consistency* of a classifier to avoid confusion. We think (1) is more useful than consistency in the context of our work.
>
> - In the first part of this work, we produced necessary and sufficient conditions for robustness. These considerations largely lose their utility under the consistency definition. For instance, "if a consistent classifier exists for a distribution p", then we can say nothing about p, because the $0$ classifier is always consistent everywhere, as the reviewer mentioned. Similarly, sufficiency is also not interesting ("for any p there always exists a consistent classifier"). The utility of studying such conditions arises from the interplay of consistency and accuracy.
>
> - (1) also nicely generalizes the notion of standard risk from classical learning theory, which can be seen by setting $\epsilon = 0$, to recover the definition of standard risk. This is not the case with consistency.
>
> - The above problems arise because $f$ can be a useless classifier, like the $0$ classifier, and be perfectly consistent. To avoid this case, one might wonder whether changing (1) to $f(\bar x) \neq f^*(x)$ makes sense, for some ground truth $f^*$. While better than consistency, we still advocate for (1), as going this route requires one to assume a perfect ground truth $f^*$, which we do not need in our theorems.
>
> In short, (1) combines consistency and accuracy, and is the appropriate object of study in our context.
>
> > Q6: In Figure 5, the blue line is 1.0 for all ε, but the certified region is not symmetric around $x$ - this doesn't make sense.
>
> Yes, our certified region $C_\lambda(x)$ is not symmetric around $x$, as you note correctly. Hence, it requires some care to compare our certificates to others existing in the literature (which are symmetric, in particular Randomized Smoothing RS). We mention these difficulties in the Experiments section around Fig 4, and outline two ways to produce a comparison, the first of which is that we compute adversarial examples for prior work which are constrained to lie on our certified set $C_\lambda(x)$, via Eq. (13). This gives us a way to empirically estimate the ProjectionRobustAccuracy, which is defined as the robust accuracy of any classifier when adversarial examples are projected onto our certificate set $C_\lambda(x)$. This is plotted in Fig 5. The "Ours Certified" line is 1.0 for all $\epsilon$ by construction, as the adversarial examples are constrained to lie on $C_\lambda(x)$, where our classifier is provably certified.
>
> To be fair to existing certified defenses, we also produce another comparison without this additional projection step, in Fig. 6 - even though our method is not tailored for symmetric certificates, we perform at par with RS. The above is a very condensed description of the experiments, and the reviewer can find more details in the text.
>
> > Q5: Discussion of computational cost.
>
> The computational complexity of obtaining the certificate $C(x)$ (Thm. 4.1) is dominated by solving the linear optimization problem (6) to obtain the set $A_\lambda$. We have $n + M$ variables in this linear program, leading to a time complexity of $\tilde{O}((n+M)^{2.38})$ [A]. It is known that the cost of solving LPs is much lower in practice [B], and it takes us $11.7$ seconds on average a single CPU for each $x$ without parallel processing.
>
> For comparison to other certified defenses (like Randomized Smoothing), we perform $T=20$ steps of Eq. (13), and each step II requires solving a quadratic optimization program, given in App. F Eq (26). This is a linearly constrained quadratic program. In practice it takes us $18.1$ seconds on average on a single CPU for each $x$ without parallel processing.
>
> Thanks for bringing this up, we will add the above to Appendix F.
>
> > Q1: Typo in L229.
>
> Indeed, we mean strongly concave. Thanks for catching that.
>
> > Q2: Adversarial perturbations do not take $x$ outside domain - is this necessary?
>
> This is a minor assumption: for instance, a small $\ell_2$ perturbation to an image of a dog cannot take it outside the domain of natural images, i.e., it will still remain a natural image. To remove this assumption, we could have required all of our classifiers to first project the input onto the domain. We found that this step introduces a lot of unnecessary clutter in the theorems, and found it cleaner to present the results with this minor assumption.
>
> > Q3: Comparison to Wang et. al. [1]
>
> In the setting of federated learning, [1] studies the task of "backdoor detection" -- find whether an "adversary provided network" $g$ differs from a "benign network" $f$ in a small, unknown set of training inputs $S$ called the backdoor. They show that detection is computationally hard in general (Prop. 1), and can also be hard given access only to a first order gradient oracle, even when $f, g$ are simple linear classifiers (Prop. 2). This work is not quite related to ours, except for the fact that they use a exponentially small volume set in the construction for Prop 2.
>
> [A]: Brand, A Deterministic Linear Program Solver in Current Matrix Multiplication Time, SODA 2020
> [B]: Spielman and Teng, Smoothed Analysis of Algorithms
>
> [9]: https://openreview.net/forum?id=JDoA6admhv&noteId=hpTXK3MMPW

---

> > ### Comment · Reviewer_eEv4 · 2023-08-12
> > **Thank you!**
> >
> > I thank the authors for their detailed response and for answering all of my questions. I am satisfied with their justification of Definition 3.1 and additional experiments. I also agree with their discussion on (1) vs consistency. I am increasing my score to 8 and recommend acceptance of this paper. I think it is well written and a good contribution to the research area. I would also ask that the authors incoroporate these additional discussions in the final manuscript.

---

### Official Review · Reviewer_8n5k · 2023-07-10

**Soundness:** 3 good
**Presentation:** 3 good
**Contribution:** 3 good
**Rating:** 7
**Confidence:** 2

**Summary:**

This paper first studies the existence problem of a robust classifier from a data concentration perspective. In particular, they prove that the (strong) concentration property of class-conditional data distributions characterizes a sufficient and necessary condition for determining whether a robust classifier exists. This seemingly contradicting result complements existing works on impossibility results of adversarially robust classification [17, 43, 11]. Built upon the assumption that the underlying distribution is concentrated on a union of low-dimensional linear subspaces, the paper proposes a new way to derive robustness certificates by leveraging techniques from sparse representation and subspace clustering. Experiments on MINST are provided as empirical evidence supporting the effectiveness of the proposed certification method.

**Strengths:**

The theoretical results are mostly well-written, and their implications are discussed in detail. I did not check the proof, but I think the results are technically correct. In addition, I very much like how this paper structures its theoretical results: a main theorem followed by a proof sketch and a detailed discussion of its implication.

The first part of this paper shows a somewhat surprising result compared with existing works deriving impossibility results of robust classification [17, 43, 11]. Assuming a robust classifier exists (i.e., a human classifier), the paper shows that the data distribution has to be concentrated in a small-volume subset in the input space. A robust classifier is guaranteed to exist if the class conditionals are both concentrated and well-separated. Given that human is robust to small, adversarial perturbations exists, these results suggest natural data distributions are likely to be concentrated, and existing impossibility results may be due to unrealistic distributional assumptions.

The second part of this paper aims to study how to leverage the underlying data structure (i.e., the fact/assumption that data are concentrated on a union of low-dimensional subspaces) for adversarially robust classification. The presented method uses the classical literature of sparse representation, which seems novel to the field of robustness certification. Overall, the paper did a good job of discussing their theoretical results and explaining their broader implications.



**Weaknesses:**

Regarding the first part of the paper, I think the presented theoretical possibility results are correct, which complements existing works [17, 43, 11]. However, I want to point out several missing references [1-4]. These works studied the concentration of measure phenomenon and its connection to robust classification/learning. There are some differences, but I believe those existing works are very relevant, thus, should be cited and discussed properly. In particular, your definition of (strongly) concentrated distribution and the notion of concentration of measure introduced in [2] seems different. Also, a key assumption imposed in them [1-4] is that there exists a ground-truth classifier (i.e., a human classifier) and a machine learning classifier is imperfect (see the definition of intrinsic robustness introduced in [3]). In other words, they assume a constant (say 1%) clean error of any machine learning classifier always exists. However, this assumption seems not to be considered in this work. Could you discuss those differences with comparisons to your work?

[1] The Relationship Between High-Dimensional Geometry and Adversarial Examples. Justin Gilmer, Luke Metz, Fartash Faghri, Samuel S. Schoenholz, Maithra Raghu, Martin Wattenberg, Ian Goodfellow. ArXiv: 1801.02774.

[2] The Curse of Concentration in Robust Learning: Evasion and Poisoning Attacks from Concentration of Measure. Saeed Mahloujifar, Dimitrios I. Diochnos, Mohammad Mahmoody. AAAI 2019

[3] Empirically Measuring Concentration: Fundamental Limits on Intrinsic Robustness. Saeed Mahloujifar, Xiao Zhang, Mohammad Mahmoody, David Evans. NeurIPS 2019

[4] Understanding Intrinsic Robustness Using Label Uncertainty. Xiao Zhang, David Evans. ICLR 2022

In addition, I am a little confused about the degree of concentration implied by Theorem 2.1. Theorem 2.1 says that if an $(\epsilon, \delta)$-robust classifier exists, all the class conditionals are $(\epsilon, K\delta)$-concentrated if they are balanced. This result seems interesting, but from my perspective, a discussion of whether this result holds for most data distributions or only applies to certain distributions is missing. Simply judging from Definition 2.1, there is little clue on what data distribution is concentrated and, more importantly, their degree of concentration. The constants $c_1$ and $c_2$ are vaguely described, so could you explain their dependence on other parameters such as the density function q and $\delta$? If the data distribution lies on a low-dimensional manifold, does it always imply the underlying distribution is concentrated?  I recommend the authors provide examples of concentrated distributions with a characterization of their degree of concentration.

The result of Theorem 3.1 is not very surprising. The assumption that the underlying class conditional distribution is strongly concentrated seems strong. To be honest, I am not convinced that this assumption will hold for natural image distributions. Otherwise, image classification would be rather a simple task. Therefore, it would be important for the authors to clarify this in the response.

Regarding the second part, I have to say I am not familiar with the literature on sparse representation. I can understand the motivation and main message of leveraging the underlying data structure, which is interesting. Still, I am concerned about the assumption that data distributions are concentrated on a union of low-dimensional linear subspaces to be too strong to hold in practice. The applicability of your method seems to be heavily dependent on whether the distribution satisfies the imposed subspace assumption. MNIST may be more aligned with such an assumption, but Gaussian-like data distributions such as CIFAR-10 and ImageNet may not be. Therefore, I am not confident that the proposed method will work for other image benchmarks like CIFAR-10.



**Questions:**

1. Could you provide a detailed discussion with related works [1-4] and explain why data concentration implies different conclusions about the possibility/impossibility of adversarially robust classification?

2. If we assume a human classifier is adversarially robust to image classification tasks, does Theorem 2.1 implies how concentrated the input distribution is? If so, what are the important factors affecting the degree of concentration?

3. Do the constants $c_1$ and $c_2$ in Definition 2.2 depend on the underlying density $q$ and $\delta$? If so, could you specify their dependencies?

4. Could you clarify the generalizability of the proposed certified defense in Section 4 if the underlying data distribution deviates from the imposed assumption regarding linear subspaces?


**Limitations:**

The paper discusses the limitations and implications of their results well in Section 6.

---

> ### Author Rebuttal · Authors · 2023-08-09
>
> Thank you for finding our paper well written and the results novel and interesting. Here are our responses to your comments:
>
> > Q1. Detailed comparison to [1-4].
>
> The results in these works support our claims. We detail several aspects below:
>
> ---
>
> **Comparison of notions of concentration**
> *Concentration* of a measure $\mu$ as considered in [2-4] denotes how much the $\mu$-measure of any set *blows up* after expansion by $\epsilon$ (i.e., over all sets $A$, with $\mu(A) \geq 0.5$ say)  what is the minimum $\mu(A^{+\epsilon})$). Under this definition, the uniform measure (in other words *volume*) has a high degree of concentration in high dimensions, and this is called the concentration of measure.
>
> In contrast, our Defn. 2.2 of concentrated data distributions can be seen as a *relative* notion of concentration with respect to the uniform measure $\mu$, in that we call a class distribution $q$ concentrated when it assigns a large $q$-measure to a set of very small volume, (i.e., $q(S)$ is high whereas $\mu(S)$ is very low).
>
> In short, we define concentration relative to the uniform distribution, whereas [2-4] define concentration independently. Our definition is useful in the context of adversarial robustness as it allows us to separate the concentration of data distributions (which is unknown) from the concentration of the uniform measure in high dimensions (for which there is a good understanding).
>
> **No need for a ground truth classifier**
> Great point! There exist two settings for robustness in the literature - the realizable setting: measuring errors against a *deterministic* ground truth classifier $f^*$, and the non-realizable setting: measuring errors against a *probabilistic* ground truth label $Y$. In the realizable setting, the best (Bayes) classifier can have a non-zero risk, and thus is strictly more general than the realizable setting. Our analysis directly handles the non-realizable setting, and thus does not require us to assume any perfect ground truth classifier.
>
> **No need for an imperfect classifier**
> As the analysis of [1-4] is conducted in the realizable setting, they need to assume a non-empty error region (say $1\%$ as the reviewer mentions) for the learnt classifier $g$ with respect to $f^*$, in order to reason about epsilon-expansions of this region. The results in [2,3] indicate that the robust risk grows fast with the volume of these error regions. However, humans seem to be a case where the natural accuracy is not perfect (e.g., we might be confused between a $6$ and a poorly written $5$ in MNIST), yet we seem to be very robust against $\ell_p$ perturbations. This points to a slack in the analysis in [2,3], and our work fills this gap by considering epsilon expansions of a different family of sets.
>
> Thanks for these useful references, we will discuss them in the paper.
>
> ---
>
> > Q3. What do the constants depend on?
>
> **Constants $c_1, c_2$, and the degree of concentration**
> The definition of concentration becomes stronger as $c_1$ becomes smaller, and as $c_2$ becomes larger. The constants are determined by the dimension of the input space $n$, and do not depend on any other variable, e.g., $q, \delta, \epsilon$. For a general distribution, we cannot say much more about the relationship between $q, \delta, \epsilon$.
>
> For Theorem 3.1, $c_1 = Vol (B_2(0, 1))$, and $c_2 = n$ (Appendix Eq.18). Here, $B_2(0,1)$ is the unit $\ell_2$ ball in $n$ dimensions, whose volume decays rapidly with $n$ as $n^{-(n + 1)/2}(2 \pi e)^{n/2}$. We will highlight the above in the paper.
>
> > Q2. Assuming robust human, what does Theorem 2.1 imply?
>
> Assume that a human classifier is adversarially robust for MNIST with robust accuracy $95\%$ at an attack strength of $0.1$ in $\ell_2$ norm. We have $\delta = 0.05$ and $\epsilon = 0.1$ for Theorem 2.1. The classes are balanced in MNIST, so that all classes are $(0.1, 0.5)$-concentrated. In words, this means that for all classes, there is a subset of the space which contains at least *half* of the class but has volume at most $c_1 \exp(-0.1 c_2)$, where the constants from the previous paragraph can be used. The factors affecting the degree of concentration are the input dimension $n$, human's robust accuracy, and the attack strength tolerated -- for each factor, higher implies stronger concentration.
>
> ---
> > Q4: How well does the method in Sec 4. generalize beyond the subspace case?
>
> Our classifier in Sec 4 can in principle be applied unchanged to any data, albeit the performance is better when the data lies near low dimensional subspaces. We provide empirical evidence for CIFAR-10 for the same in the [global response][1].
>
> ---
> **Other Concerns**
>
> > The assumption that the underlying class conditional distribution is strongly concentrated ... will [not] hold for natural image distributions, otherwise, image classification would be rather a simple task.
>
> We respectfully disagree, and argue in the [global response][1] that natural image distributions should be strongly concentrated, yet this does not make the natural image classification task trivial.
>
> > Low-dimensional support implies concentration?
>
> Yes, a manifold having dimension lower than the ambient dimension $n$ has $0$ volume, and the distribution has measure $1$ on this set, so the distribution is $(\infty, 0)$-concentrated, which is the "highest" concentration possible. This should be seen as an extreme case, our Defn 2.2 considers the natural generalization of this extreme case.
>
> > Provide examples of concentrated distributions.
>
> We have provided a few examples of concentrated distributions in Examples 3.1, 4.1 in the paper. Ex. 4.1 shows a $(\alpha, 0)$-concentrated distribution supported near a low-dimensional manifold. Ex 3.1 constructs a concentrated distribution on the unit sphere, and Appendix C (L659+) generalizes Ex 3.1. Thank you for this suggestion, we will make these examples more prominent in the text.
>
> [1]: https://openreview.net/forum?id=JDoA6admhv&noteId=hpTXK3MMPW

---

> > ### Comment · Reviewer_8n5k · 2023-08-12
> > **Thanks for the detailed response**
> >
> > I appreciate the detailed clarifications and the additional CIFAR-10 experiments. They clarified my questions and demonstrated the contribution of your work to the field. I will raise my score to reflect this, and I expect to see the above discussions integrated into the next version of your paper.

---

### Author Rebuttal · Authors · 2023-08-09

**Comment to Everybody**
We thank the reviewers for their time and thoughtful reviews. We address broader questions here, and leave the rest to individual responses. At a high level, we have explained the importance of our concentration theorems, their applicability to real world datasets, and their relationship to other work pointed out by the reviewers. On the empirical side, we have conducted new experiments on CIFAR-10, and found that the results (PDF attached) follow trends similar to our original MNIST experiments, providing further validation for our theoretical ideas. We hope to have met all the reviewers' concerns, and look forward to engaging with them during the discussion phase for any further questions.

**Are natural image distributions strongly concentrated?**
Consider the task of classifying images of dogs vs cats. Any small $\ell_2$ perturbation (say of size $\epsilon = 0.1$) to an image of a dog is not likely to change it to an image of a cat -- the true label (or a human decision) does not change with small perturbations. Let $S_{cat}$ be the set of images that are cats with a high confidence, and similarly for $S_{dog}$. What we just said was that any image in the $\epsilon$-expansion of $S_{cat}$ is not likely to be a dog, and vice versa. This is precisely the condition of separation in the definition of strong concentration: $q_{dog}(S_{cat}^{\epsilon}) \leq \gamma$, and $q_{cat}(S_{dog}^{\epsilon}) \leq \gamma$, for a small $\gamma$. The other condition of concentration is satisfied due to Theorem 2.1 assuming the existence of a robust human classifier.

Now, despite the above, the task of practically classifying an image into a dog or a cat is not trivial at all, as the sets $S_{dog}$ and $S_{cat}$ might be very complex, and hence it might be computationally hard to find a predictor (neural network, or any other classifier) for distinguishing $S_{dog}$ and $S_{cat}$. Using modern deep learning, one is able to learn an approximation to these sets that leads to small standard error, but this approximation is still bad enough that the learned predictor is very susceptible to adversarial examples. This is to say that task of robustly classifying an image into dog or cat is even harder.

**Necessity of Definition 3.1**

In Def 3.1, we are trying to obtain a sufficient condition for the existence of a robust classifier. There are two parts to Def 3.1: the concentration, and the separation conditions. The first is essential, as Theorem 2.1 says that whenever a robust classifier exists for a data distribution, the class conditionals are concentrated (i.e., it is necessary). Hence we discuss the latter:

- Firstly, as we argued above, the separation condition is not strong, and should be satisfied by image distributions like MNIST, CIFAR-10 and ImageNet.

- Secondly, let us walk through the separation condition to see whether it can be relaxed. As we note at the start of Sec. 3, if all the class conditionals were to be concentrated on subsets $S_i$ that have high intersection among each other, then even benign classification would be hard (i.e., benign risk will be high), let alone robust classification. So, even for the existence of a good benign classifier, it is essential for $S_i \cap S_j$ to be close to empty, for all $i \neq j$.

    Now, even if $S_i \cap S_j$ were almost empty, for a robust classifier, we care about the $\epsilon$-expansions of these sets to not intersect. In other words, we do not want an $\epsilon$-perturbed cat to look like a dog. Hence, for the existence of a good robust classifier, $S_i \cap S_j^{+\epsilon}$ should be close to empty, for all $i \neq j$. In measure terms, this is same as $q_i(S_j^{+\epsilon}) \leq \gamma$, for a small $\gamma$. Upto this, all these conditions are essential for obtaining a robust classifier.

    Now, note that in Def 3.1, the expansion is taken for $2 \epsilon$, instead of $\epsilon$. This is where our proof technique for Theorem 3.1 incurs a slack of $\epsilon$, and we believe a different approach for constructing the robust classifier might be able to reduce the expansion requirement to $\epsilon$. This could be an interesting avenue for relaxing Definition 3.1 and thereby strengthening Theorem 3.1, and we will add the same to the future work section.

Thus, even though Defn 3.1 leads to a robust classifier, it is really $\epsilon$-close to the minimum set of assumptions required to guarantee the existence of a robust classifier, and has not appeared in the literature so far, to the best of our knowledge.

**Experimental Results on CIFAR-10**

We have applied our method to CIFAR-10, and provided experiments in the attached rebuttal PDF. To do this, we first embed each CIFAR-10 image $x$ into a feature space $\phi(x)$, designed such that $\ell_2$-bounded perturbations in the feature space, i.e., $\phi(x) + v$, $||v||_2 \leq \epsilon$, correspond to *semantic* perturbations in the input space, e.g., distorting the image. Such a feature space $\phi$ is obtained following recent popular work in learning perceptual metrics in vision [A], where the task is precisely to learn a feature representation where $\ell_2$ distances align with human perception. We now use our subspace model on $\phi(X)$, and perform exactly the experiments in Sec 5, and make similar observations: we obtain reasonable robust accuracy using our method (Fig 1 blue lines), and this accuracy is maintained within our certified polyhedra at high epsilon, where existing defenses are not robust.

Though such experiments showing feature space certificates have appeared in the literature [B], the question of whether the perceptual representation $\phi$ is itself susceptible to small adversarial perturbations still remains for future exploration.

[A]: Zhang et. al., The Unreasonable Effectiveness of Deep Features as a Perceptual Metric, CVPR 2018
[B]: Voracek et. al., Provably Adversarially Robust Nearest Prototype Classifiers, ICML 2022

---

### Decision · Program_Chairs · 2023-09-21

**Decision:**

Accept (poster)

**Comment:**

It's nice that a somewhat implicit intuition for when adversarial examples exist has been formalized a bit more. As reviewers noted, it's unclear how practically useful the suggested certification is, as there is an issue about comparability (which the authors definitely try to address as well as possible), but the theoretical results are interesting enough even without the experiments. The writing is clear. We ask the authors to address the reviewer's suggestions in the camera-ready and perhaps in the related work section consider pointing to some more references that argue based on a similar intuition - in addition to the mentioned papers, that's e.g. the role of the margin in PAC-Bayes robust generalization bounds as in Farnia et al '18, Xiao et al. '23 and the notion of consistent perturbations as in standard vs. adversarial accuracy tradeoff paper as in Raghunathan et al. '19).